# Fleet: Few Shots Lead Effective AI-generated Image Detection

Jiaan Wang [* 1 2]  Sirui Liu [* 1 3]  Yu Li [1]  Kaiyuan Yang [1 2]  Juan Cao [1]  Sheng Tang [1]

## Abstract

AI-generated image (AIGI) detection is undergoing a critical transition from laboratory benchmarks to open-world adversarial defense. The prevalent paradigm focuses on finding static feature spaces, assuming that some invariant artifacts learned from historical data can achieve universal zero-shot generalization. While achieving saturation on several AIGI benchmarks, this static hypothesis suffers a severe performance drop against rapidly evolving generators (e.g., SD3, Nano Banana Pro). To address these limitations, we propose that the field should expand beyond "static generalization" to a new paradigm of "dynamic adaptation". We introduce **Fleet**, a framework that pioneers a dynamic paradigm of continuous few-shot evolution, enabling rapid alignment with emerging generative threats. Fleet improves few-shot adaptation by replacing unconstrained feature updates with constrained routing correction, where avoidance routing redirects novel AI samples away from Non-AI-dominated routes within decoupled subspaces. To validate this, we present **Treasure**, a benchmark spanning 64 models and 360k images, featuring diverse architectures and 20 closed-source commercial engines. Experiments reveal that while static SOTA methods fail catastrophically on modern generators, Fleet restores performance from 20.4% to 73.1% with only 10-shot adaptation on "Doubao Seedream 4.0". Code and data are available at https://github.com/ICTMCG/Fleet.

[1]Institute of Computing Technology, Chinese Academy of Sciences, Beijing, China [2]University of Chinese Academy of Sciences, Beijing, China [3]Hangzhou Institute for Advanced Study, University of Chinese Academy of Sciences, Hangzhou, China. Correspondence to: Yu Li <liyu@ict.ac.cn>.

*Proceedings of the 43rd International Conference on Machine Learning*, Seoul, South Korea. PMLR 306, 2026. Copyright 2026 by the author(s).

## 1. Introduction

AIGI detection is characterized by a strong adversarial nature driven by the rapid evolution of generative models. In the early stages, detection methods achieved remarkable success by exploiting explicit low-level artifacts, such as periodic upsampling anomalies in the frequency domain (Wang et al., 2020; Tan et al., 2024a) or structural inconsistencies rooted in neighborhood pixel relationships (Tan et al., 2024b). With the rise of diffusion models, the paradigm evolved to leverage reconstruction discrepancies (Wang et al., 2023a), intrinsic forgery subspaces (Nguyen et al., 2025; Yan et al., 2025b), or pixel-level trajectories (Zhou et al., 2026). More recently, to mitigate semantic overfitting, researchers have pivoted to harnessing the rich priors of large-scale foundation models, fusing high-level semantics with forensic features (Ojha et al., 2023; Yan et al., 2024; Tan et al., 2025; Zhang et al., 2025a; Liu et al., 2026). Crucially, despite their methodological differences, these paradigms all operate under the **"static artifact hypothesis"**: they assume that by extracting intrinsically invariant features from historical data, a detector can achieve universal *zero-shot* generalization against unseen, future generative models. Under this hypothesis, current state-of-the-art (SOTA) methods have demonstrated near-perfect performance on established benchmarks like GenImage (Zhu et al., 2023), creating a widespread perception that the task is largely solved within these closed-set environments.

However, image generation is witnessing an explosion, spanning from Generative Adversarial Networks (Goodfellow et al., 2014; Karras et al., 2021; Brock et al., 2019) to Diffusion Models (Ho et al., 2020; Dhariwal & Nichol, 2021; Esser et al., 2024) and emerging Autoregressive Transformers (**?**). This evolution introduces unpredictable artifacts that lead to significant shifts in generative distribution. Thus, the decision boundaries of static models fail to segregate novel forgery features from the seen data manifold, rendering pre-trained "invariant" features ineffective. Consequently, we observe a catastrophic performance collapse against these unseen models: as evidenced in Figure 1b, SOTA detectors that achieve saturation on GenImage plummet to near-random or even 0.12% accuracy when tested against advanced closed-source engines like Nano Banana Pro. This dramatic failure underscores a critical reality: strictly adhering to a fixed feature space is no longer viable

for open-world defense, where the adversary is continuously evolving.

To address these limitations, we propose a paradigm shift from "static generalization" to **"dynamic adaptation."** Instead of seeking a universal "silver bullet" feature, we advocate for a forensic system capable of evolving alongside the adversary. To this end, we introduce **Fleet**, a framework that pioneers a dynamic paradigm of **continuous few-shot evolution**, enabling rapid alignment with emerging generative threats. By shifting the core objective from discovering static invariants to constructing a malleable decision space, Fleet allows for precise adaptation to novel artifacts while constraining the drift of the Non-AI manifold.

Specifically, Fleet reformulates adaptation as information flow regulation via a mutually exclusive subspace routing framework. Leveraging orthogonal constraints encourages distinct routing preferences over Non-AI-oriented and AI-oriented subspaces. Crucially, Fleet employs an avoidance routing mechanism to execute a "Shunt-and-Isolate" strategy: explicitly steering novel artifacts into the forgery subspace while suppressing their activation in the Non-AI manifold. This ensures precise adaptation to emerging models while mitigating perturbation to authentic-image representations, effectively concentrating updates within the discriminative forgery subspace.

While datasets have evolved from controlled baselines like CNNDect (Wang et al., 2020) and GenImage (Zhu et al., 2023) to "in-the-wild" challenges such as WildFake (Hong et al., 2025), Chameleon (Yan et al., 2025a), and AIGIBench (Li et al., 2025b), even these contemporary benchmarks lag behind the explosion of heterogeneous and closed-source commercial engines. This lag creates a critical **evaluation blind spot** that masks the fragility of the static generalization paradigm (Figure 1a). To bridge this gap, we present **Treasure**, a frontier benchmark encompassing 360k images from 64 mainstream models, explicitly integrating 20 closed-source engines and pioneering a multi-dimensional evaluation protocol covering architecture, semantics, and artistic style. Leveraging this comprehensive benchmark, we systematically expose the generalization failure of existing SOTA AIGI detection methods and rigorously validate the efficacy of our proposed Fleet framework.

Our contributions are summarized as follows:

- **New Paradigm:** We expose the failure of the static artifact hypothesis and propose a dynamic paradigm of continuous few-shot evolution.

- **New Method:** We introduce Fleet, a subspace routing framework that leverages avoidance routing to adapt to novel forgeries efficiently while maintaining stable Non-AI representations with limited drift.

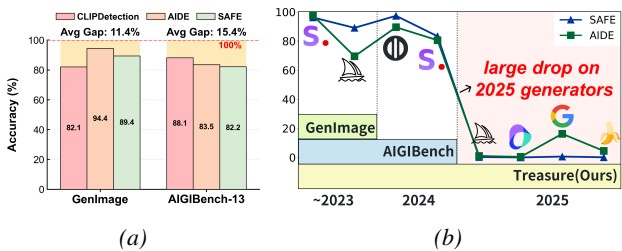

*Figure 1.* **Analysis of Benchmarks. (a)** Performance of SOTA methods on common datasets. (AIGIBench-13 refers to AI-GIBench's subset of 13 full-graph generation models.) **(b)** Detection accuracy of SOTA methods on generated images across different release years. The models in the image (from left to right) are: *SDv2.1, Midjourney v6, Playground v2.5, SD3, Midjourney v7, Doubao Seedream 3.0, Imagen 4, Nano Banana Pro.*

- **New Benchmark:** We construct **Treasure**, a benchmark integrating closed-source engines and multi-dimensional annotations to rigorously stress-test adaptation in open-world scenarios.

## 2. Related Work

### 2.1. Current Datasets for AIGI Detection

The advancement of detection methods is inextricably linked to the evolution of benchmark datasets. Early research primarily relied on datasets constructed within controlled laboratory environments. CNNDect (Wang et al., 2020) established a foundation using ProGAN (Karras et al., 2018) and LSUN (Yu et al., 2015) to test 11 architectures, though its complexity now falls behind modern generators. To bridge the widening distribution gap, GenImage (Zhu et al., 2023) constructed million-scale, category-aligned GAN/diffusion data via ImageNet prompts, while Fake2M (Lu et al., 2023) explored machine perception disparities. However, these lab-centric benchmarks often neglect the post-processing and stylistic diversity of real-world scenarios.

Consequently, research has pivoted toward "in-the-wild" challenges. WildFake (Hong et al., 2025) leverages open-source data encompassing diverse styles (e.g., anime, 3D rendering) and multi-stage pipelines. To probe detector limits, Chameleon (Yan et al., 2025a) employs human-machine collaboration to create high-difficulty samples that challenge existing SOTA models. Finally, AIGIBench (Li et al., 2025b) first decouples traditional metrics into true/false accuracy, enabling systematic evaluation of multi-source generalization capabilities and robustness.

### 2.2. Generalized AIGI Detection Methods

AIGI detection has evolved from isolated supervised learning stages to open-world generalization and rapid adaptation phases. Current approaches primarily fall into two paradigms: zero-shot detection and few-shot detection. The

distinction lies in whether target category information is leveraged during the inference process.

**Zero-Shot Methods** Early research primarily focused on CNN generators, where CNNDect (Wang et al., 2020) and FreqNet (Tan et al., 2024a) utilized periodic upsampling artifacts in spatial and frequency domains, while NPR (Tan et al., 2024b) captured structural artifacts via neighborhood pixel relationships. With the rise of diffusion models, reconstruction-based metrics became prevalent. For instance, DIRE (Wang et al., 2023a) leverages reconstruction discrepancies from pre-trained diffusion models to localize forged regions. Nguyen et al. (2025) and Effort (Yan et al., 2025b) construct intrinsic forgery subspaces through orthogonal decomposition or self-supervised filtering. Additionally, Zhou et al. (2026) proposed pixel-level mapping (PLM), which disrupts high-level semantics through preprocessing to force the model to generate generalizable pixel-level trajectories, thereby mitigating semantic overfitting.

Given the fragility of explicit artifacts, recent research has pivoted toward leveraging extensive priors from large-scale pre-trained models. CLIPDetection (Ojha et al., 2023) and C2P-CLIP (Tan et al., 2025) utilize frozen or fine-tuned CLIP encoders for classification, while AIDE (Yan et al., 2024) enhances robustness by fusing high-level CLIP semantics with low-level DCT-derived statistical features. To mitigate feature entanglement caused by task-irrelevant semantic interference, VIB-Net (Zhang et al., 2025a) and CausalCLIP (Liu et al., 2026) employ information bottlenecks or adversarial learning for feature filtration.

The zero-shot paradigm aspires to capture the invariant essence of generative models in pursuit of ultimate generalization. However, confronted with the heterogeneity of generator architectures and the leap in generation quality, such static priors prove overly idealistic and fragile against rapidly evolving "Zero-Day" threats. In contrast, few-shot learning emerges as a pragmatic compromise: by leveraging minimal priors to surmount the barriers of distribution shift, it trades minimal data costs for rapid response capabilities against unknown attacks.

**Few-Shot Methods** Cozzolino et al. (2024) demonstrates that leveraging CLIP features allows for achieving robust model generalization with only a handful of samples. FSD (Wu et al., 2025d) pioneered the incorporation of metric learning from few-shot learning into the realm of AIGI detection. Subsequently, FAMSeC (Xu et al., 2024) integrated CLIP with LoRA (Hu et al., 2022) to propose a forgery-aware module and a semantic-guided contrastive learning strategy, significantly enhancing detection robustness even with extremely limited data. Furthermore, FT-Net (Yao et al., 2026) introduced a training-free framework utilizing "failed samples"; by mining hard samples misclassified by the model to dynamically maintain a sample

gallery, it achieves rapid adaptation in few-shot scenarios.

Despite recent approaches integrating few-shot learning techniques, current AIGI detection methods suffer from architectural deficiencies and evaluation blind spots. Primarily, reliance on training-free static prototype matching fails to capture novel forgery artifacts from emerging generators, leading to prototype drift and false alarms on authentic samples. Furthermore, existing approaches often neglect catastrophic forgetting during continual adaptation, myopically focusing on short-term performance on isolated tasks. To bridge these gaps, we propose a feature fine-tuning method to effectively capture novel generative features and pioneer the introduction of an anti-forgetting metric, significantly enhancing the practical utility of few-shot AIGI detection.

## 3. Construction of Treasure

This section details the construction of Treasure, a large-scale benchmark designed to address the performance saturation and distribution lag in existing datasets. To provide a more rigorous evaluation environment for few-shot AIGI detection, this work systematically designs a diverse data collection pipeline and a multi-faceted analysis framework. The following subsections elaborate on the data acquisition process, quality control approaches, and the distinctive characteristics of the benchmark.

### 3.1. Data Collection

**Non-AI Images** Prior studies typically define "Real" as naturally captured camera images. However, the ubiquity of generative technologies in advertising and media necessitates expanding the concept opposing "Fake" to encompass all non-AI-generated content, termed **"Non-AI"** In Treasure, Non-AI images are sourced from COCO2014 and cc12m-2mp-realistic. While COCO2014 provides samples aligned with traditional "Real" distributions, cc12m contributes movie posters, illustrations, and documents to ensure diversity in semantics, artistic styles, and resolutions.

**AI Images** To enhance scale and diversity, images are curated through three distinct channels: (i) random sampling of 27 generator categories from public benchmarks and open-source communities; (ii) local generation from 21 open-source models ; and (iii) API-based generation from 16 commercial platforms. Detailed model specifications are provided in Appendix A.3. Furthermore, a comprehensive prompt library is constructed for text-to-image (T2I) sampling. Inspired by OmniDFA (Wu et al., 2025c), this library employs four complementary subsets designed to span the semantic spectrum from concise instructions to granular textual descriptions , with compositional details elaborated in Appendix A.2.

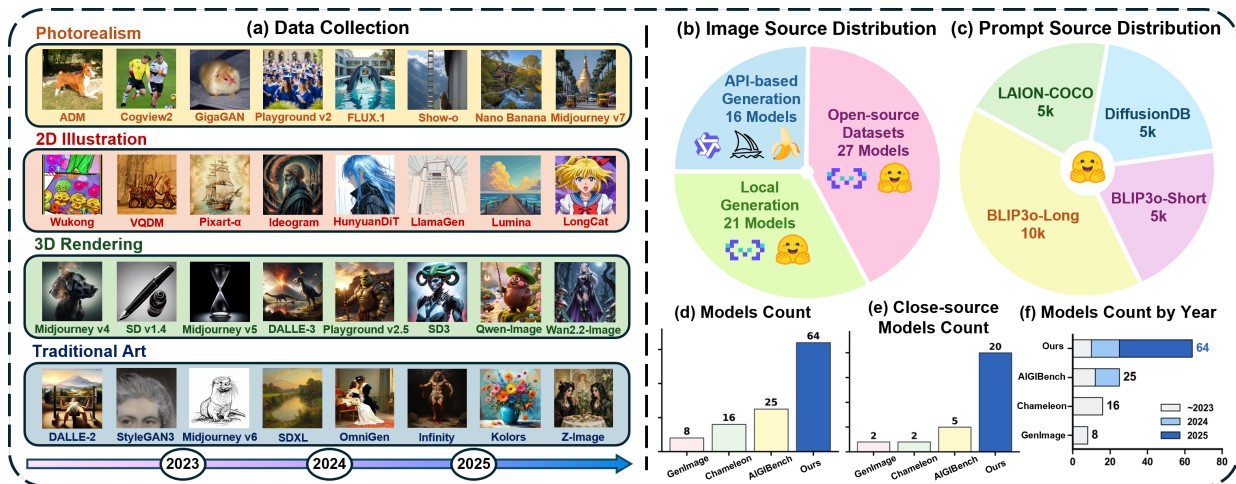

*Figure 2.* **Overview of Treasure Dataset. (a)** We have meticulously collected a large number of images generated by current mainstream generative models, assigning authenticity labels and artistic style tags to each image. **(b, c)** Illustration of image and prompt source distribution. Our dataset maintains excellent diversity in data sources to ensure its ability to simulate real-world inputs. **(d, e, f)** Comparison with other benchmarks. Our dataset possesses significant advantages in both scale and novelty.

**Quality Control** To ensure a uniform semantic distribution, feature embedding and similarity-based deduplication are applied to all images and prompts. Specifically, features are extracted using a pre-trained DINOv3 model to facilitate cosine similarity-based deduplication during image sampling. Similarly, prompt encodings are computed using the Qwen3-embedding (Zhang et al., 2025b) model, with deduplication performed on normalized feature vectors. Further filtering is executed based on prompt length and content sensitivity.

### 3.2. Analysis

The Treasure benchmark comprises 64 distinct generative categories, each containing approximately 5,000 high-resolution images, supplemented by two Non-AI image sources with 20,000 samples each, totaling about 360k images. As illustrated in Figure 2 (a), Treasure introduces novel artistic style annotations alongside standard binary and category labels. Leveraging the Qwen3-VL model, all samples are categorized into four stylistic domains—photorealism, 2D illustration, 3D rendering, and traditional art—providing a new dimension for multi-faceted generalization assessment. In summary, Treasure aggregates the most prevalent and advanced generative models, exhibiting extensive diversity in architecture, semantics, and style. It represents a highly realistic and challenging evaluation environment specifically engineered for the few-shot AIGI detection task. As shown in Figure 2 (d-f), it demonstrates significant advantages over other works in terms of diversity, scale, and novelty.

## 4. Method

In this section, we present Fleet, a mutually exclusive subspace routing method. An overview of Fleet is illustrated in Figure 3.

### 4.1. Subspace Routing Framework

Leveraging the sensitivity of high-frequency components to low-level traces (Tan et al. (2024a); Frank et al. (2020)) and the robustness of pre-trained models (Ojha et al. (2023), He et al. (2024)), we design a dual-branch architecture: the high-frequency components branch guides subspace routing, while the pre-trained branch handles feature extraction and discrimination.

**Subspace Projection** We employ the pre-trained DINOv3 (Siméoni et al., 2025) vision large model as the backbone $E_{\text{vis}}$ and specialize in its capabilities for the forgery detection task via a Low-Rank Adaptation (LoRA) matrix $\Delta\theta_{\text{lora}}$. For an input image $x$, we first extract its feature vector $\mathbf{z}$:

$$\mathbf{z} = E_{\text{vis}}(x; \theta_{\text{dino}} + \Delta\theta_{\text{lora}}) \in \mathbb{R}^D. \quad (1)$$

Here, $\mathbf{z}$ is a high-dimensional backbone representation. We use lightweight linear projections to parameterize multiple learnable subspaces. These projections generate subspace addresses for routing and subspace features for representation aggregation. Together with frequency-guided routing, orthogonality and coverage constraints, and avoidance routing during adaptation, this design promotes effective feature decoupling across AI and Non-AI responses, enabling Fleet to adapt to emerging generators while maintaining stable performance on previously learned data.

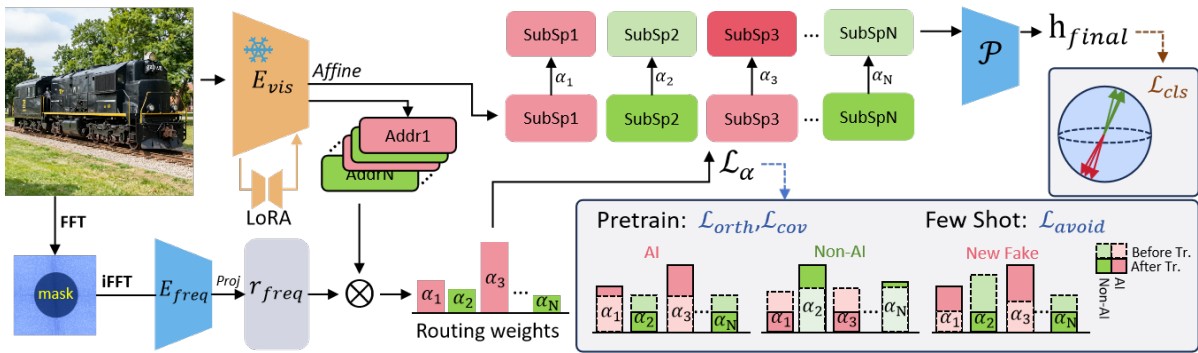

*Figure 3.* **Overview of Fleet.** The model employs a dual-branch architecture, utilizing the high-frequency signal $r_{\text{freq}}$ to generate subspace routing weights. During pre-training, $\mathcal{L}_{\text{orth}}$ and $\mathcal{L}_{\text{cov}}$ decouple high-dimensional features into mutually exclusive AI (Red) and Non-AI (Green) subspaces. In the few-shot adaptation phase, $\mathcal{L}_{\text{avoid}}$ forces the redirection of feature flows from novel generated images toward the most relevant AI subspaces while suppressing their activation in Non-AI subspaces.

To this end, we define two sets of subspace-specific linear projection parameters: $\{(\mathbf{W}_i^{\text{addr}}, \mathbf{b}_i^{\text{addr}})\}_{i=1}^C$ and $\{(\mathbf{W}_i^{\text{feat}}, \mathbf{b}_i^{\text{feat}})\}_{i=1}^C$. For the $i$-th feature subspace, its corresponding subspace address $\mathbf{a}_i$ and subspace feature $\mathbf{s}_i$ are computed via the following affine transformations:

$$
\begin{aligned}
\mathbf{a}_i &= \mathbf{W}_i^{\text{addr}}\mathbf{z} + \mathbf{b}_i^{\text{addr}}, \\
\mathbf{s}_i &= \mathbf{W}_i^{\text{feat}}\mathbf{z} + \mathbf{b}_i^{\text{feat}}, \quad i \in \{1, \ldots, C\},
\end{aligned} \tag{2}
$$

where $\mathbf{W}_i^{(\cdot)} \in \mathbb{R}^{d \times D}$ denotes the projection weight matrix, and $\mathbf{b}_i^{(\cdot)} \in \mathbb{R}^d$ is the corresponding bias vector. Specifically, $\mathbf{a}_i$ serves as the addressing signature for the $i$-th subspace, utilized for matching during the routing phase, while $\mathbf{s}_i$ represents the corresponding subspace feature to be activated. In our implementation, we set $D = 1024$ and $d = 128$.

**High-Frequency Component-Guided Subspace Routing**
To capture the low-level high-frequency features of the image, we adopt an image processing approach similar to FreqNet. First, we perform a Fast Fourier Transform (FFT) on the input image $x$ and zero out the low-frequency components to thoroughly eliminate semantic interference, preserving only the high-frequency signals sensitive to the generation mechanism:

$$
\mathbf{X}_{\text{high}} = \mathcal{F}^{-1}(\mathcal{F}(x) \odot \mathbf{M}_{\text{high}}), \tag{3}
$$

where $\mathcal{F}(\cdot)$ and $\mathcal{F}^{-1}(\cdot)$ denote the FFT and its inverse, respectively; $\mathbf{M}_{\text{high}}$ is a high-pass filter; and $\odot$ denotes element-wise multiplication. Subsequently, the processed high-frequency component $\mathbf{X}_{\text{high}}$ is fed into a convolutional encoder $E_{\text{freq}}$. Since the high-frequency component contains low-level forgery traces, $E_{\text{freq}}$ can extract semantic-agnostic high-frequency features, generating a high-frequency component-guided routing signal $\mathbf{r}_{\text{freq}}$:

$$
\mathbf{r}_{\text{freq}} = E_{\text{freq}}(\mathbf{X}_{\text{high}}) \in \mathbb{R}^d. \tag{4}
$$

We utilize the dot product between the routing signal $\mathbf{r}_{\text{freq}}$ and each subspace address $\mathbf{a}_i$ as a gating network, employ-

ing the Softmax function to generate the normalized routing distribution weights $\boldsymbol{\alpha} \in \mathbb{R}^C$:

$$
\alpha_i = \frac{\exp((\mathbf{r}_{\text{freq}}^\top \mathbf{a}_i)/\sqrt{d})}{\sum_{j=1}^C \exp((\mathbf{r}_{\text{freq}}^\top \mathbf{a}_j)/\sqrt{d})}. \tag{5}
$$

The Softmax function introduces a competitive flow control mechanism: when the avoidance mechanism forcibly suppresses the path to the Non-AI subspace, the normalization property compels the information flow of forgery features from unseen model images to automatically redirect toward the most relevant forgery subspace. The final output feature $\mathbf{h}_{\text{final}}$ is the weighted concatenation of subspace features, processed through a projection network $P(\cdot)$ for feature fusion and mapping:

$$
\mathbf{h}_{\text{final}} = P\left(\text{Concat}(\alpha_1 \mathbf{s}_1, \ldots, \alpha_C \mathbf{s}_C)\right), \tag{6}
$$

where $\text{Concat}(\cdot)$ denotes the concatenation operation along the feature dimension. The output feature $\mathbf{h}_{\text{final}}$ is then utilized to construct prototype vectors for Non-AI/AI categories for the final classification.

### 4.2. Mutually Exclusive Routing Pre-training

To ensure the effectiveness of the routing mechanism, it is imperative to enforce functional differentiation among the $C$ subspaces during the pre-training phase.

**Contrastive Learning Loss $\mathcal{L}_{\text{cls}}$** We employ the InfoNCE loss to construct a discriminative feature space:

$$
\mathcal{L}_{\text{cls}} = -\frac{1}{B} \sum_{i=1}^B \log \frac{\exp(\text{sim}(\mathbf{h}_i, \mathbf{h}_{\text{pos}})/\tau)}{\sum_{j \neq i} \exp(\text{sim}(\mathbf{h}_i, \mathbf{h}_j)/\tau)}, \tag{7}
$$

where $\mathbf{h}_i$ denotes the anchor sample, $\mathbf{h}_{\text{pos}}$ represents a positive sample sharing the same category as the anchor, and $\mathbf{h}_j$ indicates a negative sample from a different category within the batch. $B$ and $\tau$ denote batch size and temperature.

**Non-AI/AI Routing Orthogonality Constraint $\mathcal{L}_{\text{orth}}$** We compute the average routing weights for Non-AI samples, $\bar{\boldsymbol{\alpha}}_{\text{Non-AI}}$, and for AI samples, $\bar{\boldsymbol{\alpha}}_{\text{AI}}$, within a batch. By minimizing the dot product between them, we compel the subspaces to become mutually exclusive between the "Non-AI" and "AI" domains:

$$\mathcal{L}_{\text{orth}} = \sum_{c=1}^{C} \bar{\boldsymbol{\alpha}}_{\text{Non-AI}}[c] \cdot \bar{\boldsymbol{\alpha}}_{\text{AI}}[c], \quad (8)$$

where $C$ represents the number of subspaces.

**Coverage Constraint $\mathcal{L}_{\text{cov}}$** To prevent feature collapse and ensure that every subspace dimension is effectively utilized, we introduce a coverage constraint:

$$\mathcal{L}_{\text{cov}} = -\frac{1}{C} \sum_{c=1}^{C} \log(\bar{\boldsymbol{\alpha}}_{\text{Non-AI}}[c] + \bar{\boldsymbol{\alpha}}_{\text{AI}}[c] + \epsilon), \quad (9)$$

where $\epsilon$ is a small positive constant to prevent numerical instability (i.e., $\log(0)$).

**Total Pre-training Loss** The total loss during the pre-training phase is defined as:

$$\mathcal{L}_{\text{pre}} = \mathcal{L}_{\text{cls}} + \lambda_{\text{orth}}\mathcal{L}_{\text{orth}} + \lambda_{\text{cov}}\mathcal{L}_{\text{cov}}. \quad (10)$$

### 4.3. Few-Shot Adaptation via Avoidance Routing

During the adaptation phase, to address the issue where images generated by the novel model unexpectedly trigger activations in the non-AI subspace, we propose an avoidance routing strategy. Initially, we randomly select 1,000 Non-AI and AI images from the training set to construct a replay set $\mathcal{M}$, following established practices in continual learning (Rebuffi et al., 2017).

Computation of Replay Anchors. We leverage the replay set $\mathcal{M}$ to compute the fixed anchor distributions (centroids) established during the pre-training phase:

$$\boldsymbol{\mu}_{\text{Non-AI}} = \frac{1}{|\mathcal{M}_{\text{Non-AI}}|} \sum_{x \in \mathcal{M}_{\text{Non-AI}}} \boldsymbol{\alpha}(x),$$

$$\boldsymbol{\mu}_{\text{AI}} = \frac{1}{|\mathcal{M}_{\text{AI}}|} \sum_{x \in \mathcal{M}_{\text{AI}}} \boldsymbol{\alpha}(x). \quad (11)$$

**Avoidance Loss $\mathcal{L}_{\text{avoid}}$** We compel the routing distributions of novel samples within the support set $\mathcal{S}$ to "evade" incorrect anchors. Specifically, we enforce orthogonality between novel AI samples and the Non-AI anchor, as well as between Non-AI samples and the AI anchor:

$$\mathcal{L}_{\text{avoid}} = \frac{1}{|\mathcal{S}_{\text{AI}}|} \sum_{i \in \mathcal{S}_{\text{AI}}} (\boldsymbol{\alpha}_i \cdot \boldsymbol{\mu}_{\text{Non-AI}})$$
$$+ \frac{1}{|\mathcal{S}_{\text{Non-AI}}|} \sum_{j \in \mathcal{S}_{\text{Non-AI}}} (\boldsymbol{\alpha}_j \cdot \boldsymbol{\mu}_{\text{AI}}). \quad (12)$$

**Anti-Forgetting Distillation $\mathcal{L}_{\text{distill}}$** To mitigate catastrophic forgetting, we impose a distillation constraint on the replay set $\mathcal{M}$, preserving the geometric configuration of replay samples within the feature space:

$$\mathcal{L}_{\text{distill}} = \frac{1}{|\mathcal{M}|} \sum_{k \in \mathcal{M}} \Big( 1 - \text{sim}\big(\mathbf{h}_{\text{final}}^{\text{curr}}(x_k),$$
$$\mathbf{h}_{\text{final}}^{\text{old}}(x_k)\big)\Big), \quad (13)$$

where $\mathbf{h}_{\text{final}}^{\text{curr}}$ and $\mathbf{h}_{\text{final}}^{\text{old}}$ denote the features extracted by the current and previous model versions, respectively.

**Total Adaptation Loss** The total loss function for the adaptation phase is formulated as:

$$\mathcal{L}_{\text{adapt}} = \mathcal{L}_{\text{cls}} + \lambda_{\text{avoid}}\mathcal{L}_{\text{avoid}} + \lambda_{\text{distill}}\mathcal{L}_{\text{distill}}. \quad (14)$$

## 5. Experiment

### 5.1. Datasets and Evaluation Protocols

In the pre-training phase, to align with the setup used by zero-shot methods, we use the AIGIBench training set, which consists of 144k images generated by two models: ProGAN (Karras et al., 2018) and SD v1.4 (Rombach et al., 2022). In the few-shot adaptation phase, to rigorously evaluate the effectiveness of our method, we adopt two primary testing protocols: Protocol I conducts a few-shot evaluation on each category in the Treasure dataset, which encompasses 64 distinct generative models. Protocol II conducts a few-shot evaluation on each category in the public AIGIBench test set, which contains 13 testing subsets. Unlike previous works, we introduce the AIGIBench validation set as an indicator of anti-forgetting, allowing us to observe whether the few-shot adaptation process induces catastrophic forgetting. See Appendix A.1 for more details.

### 5.2. Baselines and Evaluation Metrics

We compare our proposed method against two categories of state-of-the-art (SOTA) approaches: (1) Zero-shot methods: FreqNet (Tan et al., 2024a), ClipDetection (Ojha et al., 2023), AIDE (Yan et al., 2024), SAFE (Li et al., 2025a), and PLM (Zhou et al., 2026). (2) Few-shot methods: FSD (Wu et al., 2025d). Specifically, PLM, FSD, and SAFE are reproduced based on their official code repositories, while the remaining zero-shot methods utilize their official pre-trained weights. We evaluate the accuracy of each method on both AI and Non-AI classes within the test set (query set), as well as the accuracy on the anti-forgetting set. To further assess model performance, we also employ the Average Precision (AP) metric on the test set (query set). The threshold for the accuracy metric is set to 0.5.

*Table 1.* Comparison of different methods under **Zero-shot** and **Few-shot** settings on two benchmarks: **Treasure** and **AIGIBench-13**. The reported metrics include classification accuracy (%) on the Fake Test Set, Non-AI images, and the Pretrain Validation set, as well as Mean Accuracy (mAcc) and Mean Average Precision (mAP). **Best** and second-best results are marked.

| | Method | Venue | Treasure | | | | | AIGIBench-13 | | | | |
|---|---|---|---|---|---|---|---|---|---|---|---|---|
| | | | AI | Non-AI | mAcc | mAP | Pretrain Val | AI | Non-AI | mAcc | mAP | Pretrain Val |
| ZeroShot | FreqNet | AAAI 24 | 71.87 | 75.02 | 73.42 | 77.71 | 96.00 | 89.30 | 58.09 | 73.70 | 84.38 | 96.00 |
| | ClipDetection | CVPR 23 | 72.56 | 75.77 | 74.17 | 84.33 | 96.28 | 89.54 | 72.31 | 80.93 | 90.08 | 96.28 |
| | AIDE | ICLR 25 | 70.94 | 98.56 | 84.75 | 83.72 | 98.47 | 83.47 | 87.64 | 85.56 | 91.96 | 98.47 |
| | SAFE | KDD 25 | 73.53 | 99.79 | 86.66 | 86.26 | 99.83 | 77.47 | 82.79 | 80.13 | 89.40 | 99.83 |
| | PLM | AAAI 26 | 76.56 | **99.80** | 88.18 | 90.44 | 98.96 | 83.26 | **93.88** | 88.57 | 93.55 | 98.96 |
| FewShot | FSD (1-shot) | ICML 25 | 58.51 | 71.14 | 64.82 | 68.00 | 65.38 | 59.59 | 69.14 | 64.36 | 66.31 | 63.95 |
| | FSD (5-shot) | ICML 25 | 66.56 | 79.55 | 73.05 | 77.42 | 74.15 | 69.53 | 78.45 | 73.99 | 76.80 | 74.39 |
| | FSD (10-shot) | ICML 25 | 68.28 | 83.45 | 75.87 | 80.54 | 74.81 | 71.17 | 79.45 | 75.31 | 79.47 | 76.82 |
| | **Ours (1-shot)** | – | 76.08 | 97.90 | 86.99 | 92.69 | **99.97** | 92.12 | 92.33 | 92.23 | 96.14 | **99.95** |
| | **Ours (5-shot)** | – | 84.23 | 97.50 | 90.87 | 95.26 | **99.97** | 96.01 | 92.64 | 94.33 | 97.96 | 99.80 |
| | **Ours (10-shot)** | – | **88.21** | 97.05 | **92.63** | **95.66** | 99.96 | **97.68** | 92.11 | **94.89** | **98.61** | 99.69 |

*Table 2.* Ablation study results on loss function. The table compares the performance of our method with different loss components removed. $\mathcal{L}_\alpha$ contains $\mathcal{L}_{\mathrm{orth}}, \mathcal{L}_{\mathrm{cov}}$ and $\mathcal{L}_{\mathrm{avoid}}$.

| Method | Query Set | Non-AI | Pretrain Val | **mAcc** |
|---|---|---|---|---|
| **Ours** | 91.01 | 95.51 | 99.95 | **93.26** |
| w/o $\mathcal{L}_{\mathrm{avoid}}$ | 81.60 | 96.89 | 99.97 | 89.25 |
| w/o $\mathcal{L}_{\mathrm{cov}}$ | 90.66 | 94.14 | 99.94 | 92.40 |
| w/o $\mathcal{L}_\alpha$ | 82.69 | 95.74 | 99.95 | 89.22 |
| baseline | 86.08 | 96.61 | 99.97 | 91.35 |

*Table 3.* Ablation study on the rank of LoRA modules. Results show mean accuracy.

| | AI | Non-AI | Pretrain Val | Acc |
|---|---|---|---|---|
| Rank=2 | 85.85 | 96.65 | 99.94 | 91.25 |
| Rank=4 | 91.03 | 95.21 | 99.94 | 93.12 |
| Rank=8 | 91.01 | 95.51 | 99.95 | **93.26** |
| Rank=16 | 90.64 | 93.89 | 99.98 | 92.26 |
| Rank=32 | 92.00 | 90.91 | 99.90 | 91.46 |

### 5.3. Comparisons with State-of-the-art Methods

As presented in Table 1, Fleet establishes a new state-of-the-art in few-shot AIGI detection by successfully resolving the conflict between rapid adaptability and baseline stability. **First**, our method demonstrates superior initialization capabilities. On the challenging Treasure benchmark, Fleet achieves a 1-shot accuracy of 76.08%, which not only approaches the performance of the SOTA zero-shot method PLM (76.56%) but also surpasses the 10-shot performance of the dedicated few-shot baseline FSD (68.28%). This counterintuitive result validates that our mutually exclusive routing pre-training constructs a highly structured feature space, allowing the model to align decision boundaries with minimal samples rather than learning from scratch. **Sec-**

**ond**, Fleet exhibits exceptional data utilization efficiency in high-performance regimes. Even on AIGIBench, where the baseline accuracy is already saturated above 90%—a regime typically plagued by diminishing returns—Fleet continues to convert additional support samples into significant gains, improving accuracy from 92.12% (1-shot) to 97.68% (10-shot). This proves our method's scalability in refining features without hitting early performance bottlenecks. **Finally**, we pioneer the inclusion of anti-forgetting metrics in the few-shot evaluation protocol to expose a critical vulnerability in existing methods. While FSD suffers from severe catastrophic forgetting, with accuracy on the validation set drawn from the pre-training distribution plummeting to 74.81%, Fleet leverages an effective replay mechanism to mitigate this issue and maintains near-perfect memory retention above 99.9%, ensuring robust continual adaptation without compromising historical knowledge.

### 5.4. Ablation Study

To comprehensively evaluate the effectiveness of each component, we conduct an in-depth analysis on 8 subsets of the Treasure dataset. These subsets were specifically selected because they cover diverse architectures and represent scenarios where zero-shot performance is sub-optimal.

**Loss Function Analysis** We analyze the impact of the loss functions in Table 2. Removing the avoidance loss results in a significant accuracy drop of 4.01%, indicating that the lack of explicit routing guidance causes the model to update incorrect subspaces, thereby impairing performance. Removing the coverage loss leads to a slight decrease of 0.86%, demonstrating the benefit of ensuring that every subspace is effectively utilized. Overall, removing all subspace-related design losses results in a 4.04% performance degradation, validating the necessity of our optimization objectives.

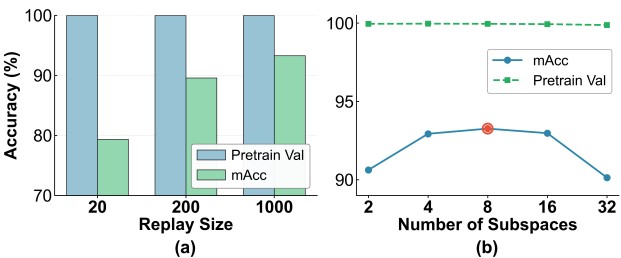

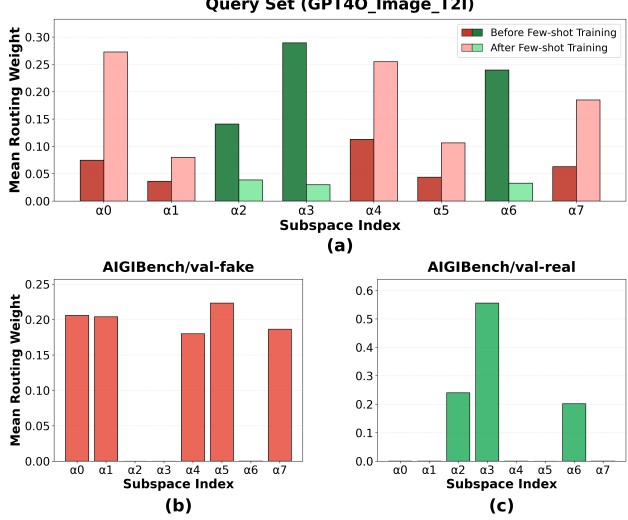

*Figure 4.* **Sensitivity Analysis.** (a) Analysis on replay buffer size. (b) Ablation on number of subspaces.

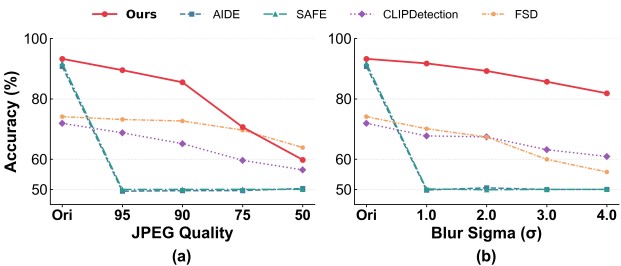

*Figure 5.* **Robust Experiment.** "Ori" means original images.

*Figure 6.* **Visualization of Subspace Routing Weights Before and After Few-shot Training.** (a) Query set mean routing weights before and after few-shot training. (b) Mean routing weights on AI classes in the AIGIBench validation set. (c) Mean routing weights on Non-AI classes in the AIGIBench validation set.

**Comparison with Baseline** To demonstrate the superiority of our architectural design, we compare our method against a standard baseline. This baseline employs DINOv3 and LoRA, fine-tuned using contrastive and distillation losses under identical replay settings. Results show that our method outperforms this general baseline by 1.91%, proving that incorporating a subspace routing mechanism adapts more effectively to forgery detection tasks compared to generic PEFT schemes.

**LoRA Rank Analysis** We also ablate the rank hyperparameter of the low-rank adaptation (LoRA) modules used during sequential adaptation. The results show that rank = 8 yields the best overall trade-off, while performance differences across a reasonable range of ranks are limited. This indicates that Fleet is not highly sensitive to this hyperparameter. Detailed results are shown in Table 3 .

**Hyperparameter Sensitivity Analysis** We analyze the impact of the replay set size. As illustrated in Figure 4, when the size is reduced to 200 and 20, accuracy drops by 3.7% and 13.92%, respectively. This performance decline is primarily attributed to insufficient data, which limits the effectiveness of contrastive learning. However, the accuracy on the pre-training validation set remains above 99.9% across all settings, demonstrating that even with a minimal replay buffer, the model maintains excellent anti-forgetting performance in forgery detection tasks. Regarding the number of subspaces, we evaluated settings of $\{2, 4, 8, 16, 32\}$. The experiments indicate that the model achieves the highest accuracy when the number of subspaces is set to 8.

### 5.5. Robustness Experiment

In real-world transmission and interaction scenarios, images inevitably encounter various unseen perturbations and degradations, posing a severe challenge to the practical deployment of AI-generated image detectors. Experimental Setup: to ensure a fair and rigorous evaluation, all comparison models are tested without any specific data augmentation or robust training. They are directly exposed to varying intensities of post-processing. Specifically, the perturbations include JPEG compression (Quality Factor $Q \in \{95, 90, 75, 50\}$) and Gaussian blur ($\sigma \in \{1.0, 2.0, 3.0, 4.0\}$). As illustrated in Figure 5, due to the absence of robust training, most baseline methods exhibit a significant performance decline. In contrast, despite these challenging and degradative conditions, our method maintains exceptional stability, significantly outperforming other approaches.

### 5.6. Visualization

Analysis of Subspace Activations. As illustrated in Figure 6, functional specialization emerges during the pre-training phase: some subspaces activate exclusively for Non-AI images, while others respond solely to AI images. However, when encountering unseen realistic AI images, subspaces associated with Non-AI data may be erroneously triggered. In response, the Avoidance Loss explicitly rectifies the routing trajectory. Consequently, as shown in Figure 6(a), following few-shot training, the activation of Non-AI-associated subspaces is effectively suppressed, whereas that of AI-associated subspaces is amplified.

*Table 4.* Results on few-shot sequential adaptation. The notation $a \rightarrow b$ denotes the accuracy before and after adaptation to the current generator. Five types of test data were sampled from the Treasure dataset.

|  | Pretrain Val | StarGAN | SDXL | GPT-4o | CogView4 | Seedream4.0 |
|---|---|---|---|---|---|---|
| After Pretrain | 99.98 | - | - | - | - | - |
| After StarGAN | 99.97 | 70.09→91.92 | - | - | - | - |
| After SDXL | 99.87 | 92.85 | 83.49→94.80 | - | - | - |
| After GPT-4o | 99.84 | 93.09 | 94.76 | 71.60→88.06 | - | - |
| After CogView4 | 99.86 | 94.01 | 94.09 | 88.46 | 79.33→83.71 | - |
| After Seedream4.0 | 99.78 | 93.00 | 94.59 | 91.04 | 84.16 | 76.47→83.53 |

*Table 5.* Sensitivity analysis of subspace initialization and addressing design. "MoE_init" refers to the structured initialization. Static $A_i$ replaces each subspace address with a fixed static prototype.

|  | AI | Non-AI | Pretrain Val | Acc |
|---|---|---|---|---|
| Fleet | 91.01 | 95.51 | 99.95 | 93.26 |
| MoE_init | 93.35 | 93.70 | 99.95 | **93.53** |
| Static $A_i$ | 88.33 | 96.09 | 99.95 | 92.21 |

### 5.7. Few-shot sequence adaptation

To further examine the behavior of our method beyond the single-step setting, we conducted a preliminary sequential adaptation experiment over five steps on previously unseen generators. The goal is to assess whether the detector can repeatedly adapt to new generators while retaining performance on earlier ones, using only a few samples per step.

The results are summarized in Table 4. After all five updates, the mean accuracy on the pretrain validation set remains 99.78% (compared to 99.98% before adaptation), indicating almost no forgetting of the original distribution. More importantly, performance on previously adapted generators is well maintained or even improves. For example, the accuracy on GPT-4o rises from 88.06% after its initial adaptation to 91.04% after subsequent updates on later generators. Similar trends are observed for other models (e.g., StarGAN, SDXL, CogView4, Seedream4.0). These results suggest that our method has promising potential for sequential adaptation beyond the single-step scenario, and we plan to investigate this direction more thoroughly in future work.

### 5.8. Analysis of Initialization and Routing Stability

We analyze two design choices in Fleet's subspace mechanism: the initialization of subspace parameters, and whether the subspace address is input-dependent (dynamic) or static. Although these are distinct aspects, both reflect degrees of freedom in how subspaces are constructed and selected. For initialization, we compare the default random initialization with a structured MoE-style warm start inspired by partial re-initialization (Nakamura et al., 2025), where subspace projections are initialized through a pretrained low-dimensional projection layer. This leads to only a marginal change in accuracy (93.26% → 93.53%), while pretrain-validation accuracy remains unchanged (99.95% → 99.95%), indicat-

ing that Fleet does not rely on special initialization tricks. For addressing, replacing the input-dependent dynamic address with a fixed static prototype per subspace reduces overall accuracy from 93.26% to 92.21%, with the main drop occurring on the AI class. This shows that dynamic addressing is beneficial for modeling emerging generator artifacts. Fleet maintains routing stability during both pre-training and adaptation. $\mathcal{L}_{\text{orth}}$ encourages distinct routing patterns for AI and non-AI samples, while $\mathcal{L}_{\text{avoid}}$ prevents routing collapse. During adaptation, the model prioritises routing correction for the first five epochs before enabling the classification term, which reduces potential optimization conflicts. Detailed results are shown in Table 5.

## 6. Conclusion and Limitations

This work introduces Treasure, a comprehensive benchmark, and Fleet, a frequency-guided subspace routing framework, to address the performance collapse of AIGI detectors when facing emerging generators. By establishing a high-margin evaluation environment with 64 generative models, this work fills an important gap in benchmark coverage. Experimental results demonstrate that Fleet effectively mitigates feature submersion through mutually exclusive routing, achieving SOTA performance across multiple benchmarks while maintaining robustness against catastrophic forgetting. These contributions establish a practical and efficient paradigm for securing digital media against the rapid evolution of synthesis technologies.

Despite the advancements, several limitations remain. First, this research focuses exclusively on full-image generation; the generalizability of the proposed method to localized editing or face-swapping tasks requires further investigation. Second, the current paradigm is designed for single-step few-shot adaptation, whereas exploring continual few-shot learning represents a critical step toward practical deployment and will be a focus of future work. Finally, the reliance on a replay set to balance performance and knowledge retention incurs additional storage overhead, suggesting that the development of more efficiency-oriented, replay-free adaptation mechanisms remains a significant objective for subsequent research. Our future work will further explore these limitations.

## Acknowledgements

This work was supported by the Institute Innovation Project of the Institute of Computing Technology, Chinese Academy of Sciences under Grant No. E561090.

## Impact Statement

This research advances AI-generated image forensics to safeguard digital integrity and curb misinformation. By improving detector adaptability, this work strengthens defenses against evolving generative models and restores public trust in media. While forensic progress may incite an "arms race" regarding forgery sophistication, comprehensive benchmarks facilitate rigorous community evaluation. All efforts prioritize ethical standards and privacy regulations to ensure positive societal impact.

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

# A. Appendix

## A.1. Experimental Details

### A.1.1. MODEL ARCHITECTURE

We utilize DINOv3 as the backbone architecture. During the training process, the backbone parameters of DINOv3 remain frozen. We employ Low-Rank Adaptation (LoRA) with a rank of 8 and a scaling factor of 16. For the high-frequency component feature extraction module ($E_{\text{freq}}$), we adopt the Xception model as the backbone and apply a high-pass filter to retain the top 50% of high-frequency components. A temperature parameter $\tau = 0.07$ is used in the contrastive learning.

### A.1.2. PRE-TRAINING PHASE

The model is trained for 3 epochs with a batch size of 256 and a learning rate of $1 \times 10^{-4}$. The loss weights are set as $\lambda_{\text{cls}} = 1$, $\lambda_{\text{orth}} = 0.1$, and $\lambda_{\text{cov}} = 10^{-4}$.

### A.1.3. FEW-SHOT ADAPTATION PHASE

Replay Set Construction: The replay set consists of 1,000 images drawn from the pre-training set, comprising 500 randomly selected AI-generated images and 500 Non-AI images. Support Set Source: To better align the distribution of non-AI images in the support set with that of AI images during adaptation, all non-AI images in the support set are sourced from the cc12m-2mp-realistic dataset. Training Configuration: Each training batch consists of two parts: all samples from the current support set (permanently included) plus 32 images alternately drawn from the replay set. The learning rate is set to $3 \times 10^{-5}$, and training is conducted for 20 epochs. Loss Strategy: The base loss weights are $\lambda_{\text{cls}} = 5$, $\lambda_{\text{distill}} = 10$, and $\lambda_{\text{avoid}} = 20$. However, in the first 5 epochs, we set $\lambda_{\text{cls}} = 0$ to prioritize learning the correct routing path.

### A.1.4. PREPROCESSING & HARDWARE

The high-frequency component branch employs random cropping during training and center cropping during testing. All experiments are conducted on 8 NVIDIA GeForce RTX 3090 GPUs. Under this setup, pre-training requires approximately 3 hours, while the adaptation phase for each subset takes about 30 minutes.

### A.1.5. DATASETS & EVALUATION SCOPE

**AIGIBench-13.** In the main experiment, "AIGIBench-13" refers to a subset containing 13 whole-image generation datasets, specifically: ProGAN, R3GAN, StyleGAN3, StyleGAN-XL, StyleSwim, WFIR, DALLE-3, FLUX1-dev, GLIDE, Imagen3, Midjourney, SD3, and SDXL.

**Ablation Study.** The ablation study covers eight representative datasets to encompass a range of mainstream methodologies (pixel-space diffusion, GPT-based, flow matching, unified GANs, autoregressive, and multimodal diffusion transformers). These include: ADM, GPT-4o, FLUX.2, StarGAN, NextStep, Qwen-Image, HunyuanImage 3.0, and wan2.5-t2i-preview.

## A.2. Collection of Prompts

Prompts serve as a critical determinant of the semantic distribution in generated imagery. To simulate the multifaceted generation requests encountered in real-world scenarios, simple category labels are eschewed in favor of a comprehensive library containing 25,000 candidate prompts. This library is structured into four complementary subsets, spanning the semantic spectrum from concise instructions to elaborate textual descriptions:

- Authentic User Input: Consists of 5,000 real-world prompts sourced from DiffusionDB (Wang et al., 2023b). This subset preserves characteristic user behaviors, such as idiosyncratic spelling, modifier stacking, and non-standard syntax, ensuring that synthesized content reflects actual application distributions.

- Long-form Descriptions: Includes 5,000 fine-grained descriptions generated by Blip3o-long (BLIP3o, 2025a), averaging 120 tokens. These prompts are designed to evaluate the generative model's capacity for rendering intricate spatial relationships and detailed textures.

- Short-form Instructions: Comprises 10,000 concise descriptions from Blip3o-short (BLIP3o, 2025b), with an average length of 20 tokens, facilitating the assessment of model performance under sparse semantic constraints.

- Open-world Metadata: Contains 5,000 high-aesthetic-score texts from LAION-COCO-Aesthetic (guangyil, 2025). Focusing on captions, slogans, and stylistic descriptors, this subset enhances coverage across non-natural image domains, such as posters and UI designs.

We provide some examples in Table 6.

*Table 6.* Exemplar prompts from the four subsets of the Treasure library.

| Subset | Exemplar Prompts |
|---|---|
| **Authentic User Input** | (1) *Emma Watson as migrant mother, 1936 photo by Dorothea Lange*
(2) *fantasy character portrait photo. female dwarf. short, broad, extremely muscular, broad face resembles cara delevingne but very squat, elaborately braided orangepink hair.*
(3) *zombie storm trooper highly detailed 1970s horror star wars art* |
| **Long-form Descriptions** | (1) *The image depicts a large, open plaza in front of a prominent building with a distinctive dome and columns, resembling a government or cultural institution. The plaza is paved and features several statues, including one of a group of soldiers and another of an individual. A colorful red and yellow carpet is laid out near the center, possibly for a ceremonial event. In the background, modern high-rise buildings are visible under a clear sky.*
(2) *The image shows a close-up of a large, muddy tractor tire and part of the tractor's body. The tractor is red with visible wear and tear, indicating it has been used in rugged conditions. The tire is heavily caked with mud, suggesting recent off-road activity. The background features a grassy area with trees, hinting at a rural or forested setting.*
(3) *The image shows a large truck driving on a concrete bridge. The truck is carrying a green trailer with the text "SUPERMERCATI BASKO" prominently displayed in red and yellow letters. The bridge appears to be part of a highway or major road, supported by large concrete pillars. The sky above is partly cloudy, suggesting it might be a cool or overcast day. The scene captures a moment of transportation, possibly during a routine journey. The overall setting indicates a modern infrastructure environment.* |
| **Short-form Instructions** | (1) *A clean workspace setup with an HP monitor, keyboard, and mouse on a wooden desk.*
(2) *Modern conference room setup with tables, chairs, and water bottles.*
(3) *Silhouetted Egypt: Pyramids, Sphinx, and Camels in a Desert Scene.* |
| **Open-world Metadata** | (1) *CASE ELEGANCE Monogrammed Full Grain Premium Leather Refillable Journal Cover with A5 Lined Notebook, Pen Loop, Card Slots, Brass Snap*
(2) *Surya Contemporary Square pouf/ottoman 24"x24"x13" in Purple Color From Surya Poufs Collection*
(3) *Apple & Cinnamon Baked Oatmeal Pie* |

## A.3. Collection of Non-AI/Fake Images

Tables 7 and 8 respectively display the **source** and **link** information for Non-AI and Fake images in Treasure dataset.

*Table 7.* Sources of Non-AI images used in Treasure.

| ID | Dataset | Source | Link |
|---|---|---|---|
| 1 | COCO2014 | Huggingface | `https://huggingface.co/datasets/AbdoTW/COCO_2014` |
| 2 | cc12m-2mp-realistic | Huggingface | `https://huggingface.co/datasets/opendiffusionai/cc12m-2mp-realistic` |

*Table 8.* Sources of generative models used in Treasure.

| ID | Model | Source | Link |
|---|---|---|---|
| 1 | BigGAN (Brock et al., 2019) | GenImage | `https://github.com/GenImage-Dataset/GenImage` |
| 2 | ADM (Dhariwal & Nichol, 2021) | GenImage | `https://github.com/GenImage-Dataset/GenImage` |
| 3 | GLIDE (Nichol et al., 2022) | GenImage | `https://github.com/GenImage-Dataset/GenImage` |
| 4 | Wukong (MindSpore Community, 2022) | GenImage | `https://github.com/GenImage-Dataset/GenImage` |
| 5 | VQDM (Gu et al., 2022) | GenImage | `https://github.com/GenImage-Dataset/GenImage` |
| 6 | SD v1.4 (Rombach et al., 2022) | GenImage | `https://github.com/GenImage-Dataset/GenImage` |
| 7 | SD v1.5 (Rombach et al., 2022) | GenImage | `https://github.com/GenImage-Dataset/GenImage` |
| 8 | Midjourney V5 | GenImage | `https://github.com/GenImage-Dataset/GenImage` |
| 9 | ProGAN (Karras et al., 2018) | WildFake | `https://github.com/hy-zpg/AIGC-Image-Detection-Dataset` |
| 10 | StarGAN (Choi et al., 2018) | WildFake | `https://github.com/hy-zpg/AIGC-Image-Detection-Dataset` |
| 11 | DF-GAN (Tao et al., 2022) | WildFake | `https://github.com/hy-zpg/AIGC-Image-Detection-Dataset` |
| 12 | StyleGAN3 (Karras et al., 2021) | WildFake | `https://github.com/hy-zpg/AIGC-Image-Detection-Dataset` |
| 13 | DALL·E 2 (Ramesh et al., 2022) | WildFake | `https://github.com/hy-zpg/AIGC-Image-Detection-Dataset` |
| 14 | Imagen (Saharia et al., 2022) | WildFake | `https://github.com/hy-zpg/AIGC-Image-Detection-Dataset` |
| 15 | Midjourney V4 | WildFake | `https://github.com/hy-zpg/AIGC-Image-Detection-Dataset` |

| ID | Model | Source | Link |
|---|---|---|---|
| 16 | MAE (He et al., 2022) | WildFake | https://github.com/hy-zpg/AIGC-Image-Detection-Dataset |
| 17 | GigaGAN (Kang et al., 2023) | WildFake | https://github.com/hy-zpg/AIGC-Image-Detection-Dataset |
| 18 | SDXL (Podell et al., 2024) | WildFake | https://github.com/hy-zpg/AIGC-Image-Detection-Dataset |
| 19 | CogView2 (Ding et al., 2021) | MPBench | https://huggingface.co/datasets/InfImagine/FakeImageDataset |
| 20 | SD v2.1 (Stability AI, 2022) | MPBench | https://huggingface.co/datasets/InfImagine/FakeImageDataset |
| 21 | DeepFloyd IF (DeepFloyd, 2023) | MPBench | https://huggingface.co/datasets/InfImagine/FakeImageDataset |
| 22 | Ideogram (Ideogram AI, 2024) | HuggingFace | https://huggingface.co/datasets/terminusresearch/ideogram-75k |
| 23 | PixArt-$\alpha$ (Chen et al., 2024) | HuggingFace | https://huggingface.co/datasets/PixArt-alpha/PixArt-Eval30K |
| 24 | DALL·E 3 (OpenAI, 2023) | HuggingFace | https://huggingface.co/datasets/OpenDatasets/dalle-3-dataset |
| 25 | FLUX.1-dev (Black Forest Labs, 2024) | HuggingFace | https://huggingface.co/datasets/lehduong/flux_generated |
| 26 | Midjourney V6 | HuggingFace | https://huggingface.co/datasets/terminusresearch/midjourney-v6-520k-raw |
| 27 | GPT-4o (OpenAI, 2024) | HuggingFace | https://huggingface.co/datasets/yufan/GPT4O_Image_T2I |
| 28 | Playground V2 (Li et al.) | Self-generated | https://huggingface.co/playgroundai/playground-v2-1024px-aesthetic |
| 29 | Playground V2.5 (Li et al., 2024a) | Self-generated | https://huggingface.co/playgroundai/playground-v2.5-1024px-aesthetic |
| 30 | Hunyuan-DiT (Li et al., 2024b) | Self-generated | https://huggingface.co/Tencent-Hunyuan/HunyuanDiT-v1.2-Diffusers |
| 31 | LlamaGen (Sun et al., 2024) | Self-generated | https://github.com/FoundationVision/LlamaGen |
| 32 | SD3-medium (Esser et al., 2024) | Self-generated | https://huggingface.co/stabilityai/stable-diffusion-3-medium |

| ID | Model | Source | Link |
|---|---|---|---|
| 33 | Show-o (Xie et al., 2025b) | Self-generated | https://github.com/showlab/Show-o |
| 34 | OmniGen (Xiao et al., 2025) | Self-generated | https://huggingface.co/Shitao/OmniGen-v1 |
| 35 | CogView3plus (?) | Self-generated | https://github.com/zai-org/CogView4 |
| 36 | Infinity-2B (Han et al., 2025b) | Self-generated | https://github.com/FoundationVision/Infinity |
| 37 | Janus-Pro-7B (Chen et al., 2025) | Self-generated | https://github.com/deepseek-ai/Janus |
| 38 | SANA v1.5 (Xie et al., 2025a) | Self-generated | https://huggingface.co/Efficient-Large-Model/SANA1.5_4.8B_1024px_diffusers |
| 39 | LUMINA-Image 2.0 (Qin et al., 2025) | Self-generated | https://huggingface.co/Alpha-VLLM/Lumina-Image-2.0 |
| 40 | HiDream-I1-Dev (Cai et al., 2025b) | Self-generated | https://github.com/HiDream-ai/HiDream-I1 |
| 41 | BAGEL (Deng et al., 2025) | Self-generated | https://github.com/ByteDance-Seed/Bagel |
| 42 | BRIA 3.2 (BRIA AI, 2024) | Self-generated | https://huggingface.co/briaai/BRIA-3.2 |
| 43 | OmniGen2 (Wu et al., 2025b) | Self-generated | https://github.com/VectorSpaceLab/OmniGen2 |
| 44 | Show-o2 (Xie et al., 2025c) | Self-generated | https://github.com/showlab/Show-o/tree/main/show-o2 |
| 45 | Ovis-U1 (Wang et al., 2025) | Self-generated | https://github.com/AIDC-AI/Ovis-U1 |
| 46 | NextStep-1 (Han et al., 2025a) | Self-generated | https://huggingface.co/stepfun-ai/NextStep-1-Large |
| 47 | Z-Image-Turbo (Cai et al., 2025a) | Self-generated | https://modelscope.cn/models/Tongyi-MAI/Z-Image-Turbo |
| 48 | LongCat-Image (Meituan LongCat Team et al., 2025) | Self-generated | https://github.com/meituan-longcat/LongCat-Image |
| 49 | Kolors (Kolors Team, 2024) | API | https://kolors-ai.com/ |
| 50 | Qwen-Image (Wu et al., 2025a) | API | https://www.aliyun.com/ |
| 51 | Imagen 4 (Fortin et al., 2025) | API | https://ai.google.dev/gemini-api |
| 52 | Nano Banana | API | https://ai.google.dev/gemini-api |
| 53 | Nano Banana Pro | API | https://ai.google.dev/gemini-api |
| 54 | Doubao Seedream 4.0 (Team Seedream et al., 2025) | API | https://seedream-4.io/ |
| 55 | Doubao Seedream 3.0 (Gao et al., 2025) | API | https://console.volcengine.com/ |
| 56 | HunyuanImage 3.0 (Cao et al., 2025) | API | https://cloud.tencent.com/ |

| ID | Model | Source | Link |
|---|---|---|---|
| 57 | FLUX.2 (Labs, 2025) | API | https://flux-2.studio/ |
| 58 | wan2.2-t2i-flash (Team Wan et al., 2025) | API | https://www.aliyun.com/ |
| 59 | wan2.5-t2i-preview (Team Wan et al., 2025) | API | https://www.aliyun.com/ |
| 60 | CogView4 | API | https://bigmodel.cn/ |
| 61 | Sora-image | API | https://sora.chatgpt.com/ |
| 62 | GPT-image-1.5 (OpenAI, 2025) | API | https://chatgpt.com/images/ |
| 63 | Midjourney v6.1 | API | https://www.midjourney.com/ |
| 64 | Midjourney v7 | API | https://www.midjourney.com/ |

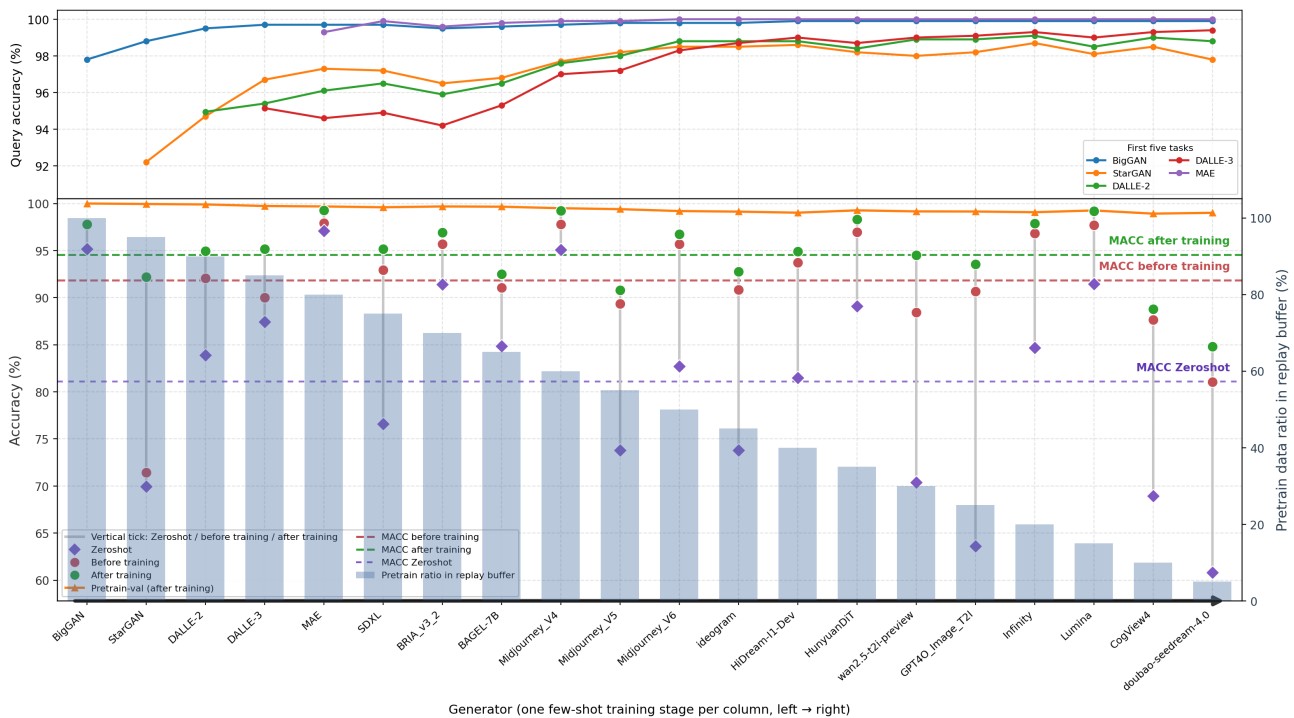

*Figure 7.* Long-horizon continual few-shot adaptation over 20 sequential generator stages. The upper panel tracks the query accuracy of the first five adapted generators during later stages. The lower panel shows zero-shot, before-training, and after-training accuracy for each newly introduced generator, along with the pretraining validation accuracy and the decreasing pretraining-data ratio in the replay buffer.

## A.4. Long-horizon continual few-shot adaptation.

We further evaluate Fleet under a longer sequential adaptation setting to study whether limited replay data for earlier generators can cause retention or adaptation degradation. Specifically, we construct a 20-stage generator sequence, where each stage introduces one new generator with few-shot supervision. Since the replay buffer has a fixed size, the relative proportion of samples from the original pretraining distribution gradually decreases as more adapted generators are included, making later stages more challenging.

As shown in Fig. 7, Fleet remains stable over the full sequence. The pretraining validation accuracy stays close to its initial level across all 20 stages, indicating that the model does not forget the original training distribution. The first five adapted generators also keep high query accuracy after later adaptations, showing that early new-class knowledge is not overwritten.

For each newly introduced generator, few-shot adaptation consistently improves the query accuracy over both the zero-shot result and the before-training result. Although later stages may benefit from accumulated exposure to a broader set of generators, the main observation is that Fleet does not collapse under this longer and more constrained replay setting. This suggests that the proposed framework can support long-horizon deployment beyond the short sequential setting used in the main experiments.

### A.5. More comparisons between Treasure and other datasets

Table 9 presents further comparisons between our dataset and other datasets, demonstrating that our work shows significant advantages in terms of model quantity, novelty, semantic diversity, and stylistic diversity.

*Table 9.* Comparison of benchmarks for AI-generated images detection.

| Benchmark | Venue | Generate Models | ~23 | 24 | 25 | Semantics Diversity | Styles Diversity | Close-Source |
|-----------|-------|-----------------|-----|----|----|---------------------|------------------|--------------|
| CNNDect | CVPR '20 | 11 | 12 | 0 | 0 | ✗ | ✗ | 0 |
| Genimage | NeurIPS '23 | 8 | 8 | 0 | 0 | ✗ | ✗ | 2 |
| MPBench | NeurIPS '23 | 11 | 11 | 0 | 0 | ✓ | ✗ | 1 |
| WildFake | AAAI '25 | 22 | 22 | 0 | 0 | ✓ | ✓ | 5 |
| Chameleon | ICLR '25 | 16 | 16 | 0 | 0 | ✓ | ✓ | 2 |
| AIGIBench | NeurIPS '25 | 25 | 14 | 11 | 0 | ✓ | ✓ | 5 |
| **Treasure (Ours)** | – | **64** | **26** | **9** | **29** | ✓ | ✓ | **20** |

## A.6. Detailed Results of Comparative Experiments

In this section, we provide a detailed breakdown of Table 1. Table 10 and Table 12 present the accuracies of various methods on AI-generated and non-AI images for each subset of the AIGIBench-13 and Treasure datasets, respectively. Table 11 and Table 13 report the accuracies on the pre-training validation set after the few-shot methods are fine-tuned separately on each subset of AIGIbench-13 and Treasure, respectively.

*Table 10.* Detailed results on AIGIbench-13.

| Setting | Method | ProGAN | | R3GAN | | StyleGAN3 | | StyleGAN-XL | | StyleSwim | | WFIR | |
|---|---|---|---|---|---|---|---|---|---|---|---|---|---|
| | | AI | Non-AI | AI | Non-AI | AI | Non-AI | AI | Non-AI | AI | Non-AI | AI | Non-AI |
| ZeroShot | FreqNet | 99.3 | 98.4 | 69.5 | 60.0 | 98.4 | 64.8 | 97.8 | 59.7 | 98.0 | 59.8 | 97.3 | 22.7 |
| | ClipDetection | 98.7 | 98.9 | 95.3 | 71.6 | 87.5 | 77.8 | 98.2 | 70.8 | 99.3 | 73.6 | 96.2 | 49.3 |
| | AIDE | 95.4 | 99.1 | 99.1 | 86.7 | 91.2 | 85.0 | 91.7 | 85.9 | 82.2 | 85.4 | 41.5 | 99.8 |
| | SAFE | 99.5 | 100.0 | 92.5 | 81.9 | 84.8 | 81.6 | 71.6 | 81.0 | 100.0 | 80.7 | 7.9 | 100.0 |
| | PLM | 98.6 | 99.5 | 98.1 | 93.7 | 95.5 | 93.2 | 99.8 | 92.6 | 100.0 | 93.4 | 0.0 | 100.0 |
| FewShot | FSD (10-shot) | 96.0 | 99.3 | 76.0 | 58.9 | 69.9 | 86.5 | 70.4 | 75.1 | 69.5 | 79.6 | 57.2 | 56.1 |
| | **Ours (10-shot)** | 100.0 | 99.3 | 99.2 | 93.5 | 97.9 | 92.3 | 94.7 | 94.6 | 98.6 | 91.7 | 97.8 | 76.3 |

| Method | DALLE-3 | | FLUX1-dev | | GLIDE | | Imagen3 | | Midjourney | | SD3 | | SDXL | |
|---|---|---|---|---|---|---|---|---|---|---|---|---|---|
| | AI | Non-AI | AI | Non-AI | AI | Non-AI | AI | Non-AI | AI | Non-AI | AI | Non-AI | AI | Non-AI |
| FreqNet | 73.7 | 59.5 | 91.3 | 59.7 | 82.6 | 67.7 | 81.1 | 61.3 | 85.2 | 20.0 | 88.1 | 60.8 | 98.7 | 60.8 |
| ClipDetection | 77.9 | 73.4 | 82.1 | 72.1 | 80.2 | 77.6 | 82.0 | 72.0 | 76.7 | 49.4 | 95.4 | 76.7 | 94.5 | 76.7 |
| AIDE | 24.7 | 85.5 | 90.1 | 86.0 | 98.4 | 88.6 | 94.0 | 85.7 | 80.0 | 72.9 | 99.3 | 89.3 | 97.6 | 89.3 |
| SAFE | 0.4 | 81.3 | 95.7 | 81.2 | 95.0 | 87.6 | 81.0 | 81.7 | 91.6 | 48.2 | 87.3 | 85.5 | 99.8 | 85.5 |
| PLM | 0.2 | 93.1 | 99.0 | 92.8 | 99.6 | 95.7 | 97.6 | 93.2 | 96.3 | 83.2 | 98.0 | 95.0 | 99.8 | 95.0 |
| FSD (10-shot) | 71.2 | 84.1 | 72.4 | 88.0 | 71.3 | 84.5 | 70.3 | 74.9 | 70.3 | 82.8 | 63.3 | 79.1 | 67.5 | 83.7 |
| **Ours (10-shot)** | 98.7 | 90.7 | 94.3 | 98.7 | 96.0 | 95.8 | 92.3 | 89.6 | 97.3 | 93.8 | 99.7 | 94.3 | 100.0 | 93.4 |

*Table 11.* Few-shot performance on the Pretrain Validation set of AIGIbench-13.

| Method | ProGAN | | R3GAN | | StyleGAN3 | | StyleGAN-XL | | StyleSwim | | WFIR | |
|---|---|---|---|---|---|---|---|---|---|---|---|---|
| | AI | Non-AI | AI | Non-AI | AI | Non-AI | AI | Non-AI | AI | Non-AI | AI | Non-AI |
| FSD (10-shot) | 50.1 | 99.5 | 60.0 | 86.2 | 93.2 | 95.3 | 66.0 | 97.8 | 95.8 | 96.6 | 63.6 | 82.7 |
| **Ours (10-shot)** | 100.0 | 99.3 | 100.0 | 99.3 | 100.0 | 99.4 | 100.0 | 99.6 | 100.0 | 99.4 | 100.0 | 99.0 |

| Method | DALLE-3 | | FLUX1-dev | | GLIDE | | Imagen3 | | Midjourney | | SD3 | | SDXL | |
|---|---|---|---|---|---|---|---|---|---|---|---|---|---|
| | AI | Non-AI | AI | Non-AI | AI | Non-AI | AI | Non-AI | AI | Non-AI | AI | Non-AI | AI | Non-AI |
| FSD (10-shot) | 57.8 | 94.6 | 62.4 | 98.1 | 91.9 | 82.0 | 45.7 | 61.4 | 47.5 | 36.0 | 68.6 | 94.1 | 73.1 | 97.3 |
| **Ours (10-shot)** | 100.0 | 99.8 | 100.0 | 99.9 | 100.0 | 99.6 | 100.0 | 99.0 | 100.0 | 99.6 | 100.0 | 99.3 | 100.0 | 99.3 |

*Table 12.* Detailed results on Treasure.

| Setting | Method | ADM | | BAGEL | | BRIA v3.2 | | BigGAN | | CogView2 | | CogView4 | |
|---|---|---|---|---|---|---|---|---|---|---|---|---|---|
| | | AI | Non-AI | AI | Non-AI | AI | Non-AI | AI | Non-AI | AI | Non-AI | AI | Non-AI |
| ZeroShot | FreqNet | 32.7 | 75.0 | 82.9 | 75.0 | 84.4 | 75.0 | 97.9 | 75.0 | 65.5 | 75.0 | 21.7 | 75.0 |
| | CLIPDetection | 14.3 | 75.8 | 73.3 | 75.8 | 83.3 | 75.8 | 77.0 | 75.8 | 90.5 | 75.8 | 46.7 | 75.8 |
| | AIDE | 92.9 | 98.6 | 78.9 | 98.6 | 80.0 | 98.6 | 73.8 | 98.6 | 11.8 | 98.6 | 0.7 | 98.6 |
| | SAFE | 62.4 | 99.8 | 96.8 | 99.8 | 98.2 | 99.8 | 0.0 | 99.8 | 55.7 | 99.8 | 0.1 | 99.8 |
| | PLM | 98.2 | 99.8 | 98.2 | 99.8 | 98.7 | 99.9 | 97.3 | 99.7 | 1.4 | 99.9 | 0.0 | 99.9 |
| FewShot | FSD (10-shot) | 75.7 | 47.8 | 73.3 | 94.0 | 75.3 | 91.3 | 79.0 | 80.7 | 71.7 | 95.0 | 63.0 | 81.7 |
| | **Ours (10-shot)** | 85.7 | 92.8 | 80.4 | 98.3 | 92.1 | 99.0 | 94.8 | 99.8 | 98.6 | 99.8 | 78.1 | 97.4 |

| Method | Cogview3plus | | DALL·E 2 | | DALL·E 3 | | DF-GAN | | DeepFloyd IF | | FLUX.1-dev | | FLUX.2 | |
|---|---|---|---|---|---|---|---|---|---|---|---|---|---|---|
| | AI | Non-AI | AI | Non-AI | AI | Non-AI | AI | Non-AI | AI | Non-AI | AI | Non-AI | AI | Non-AI |
| FreqNet | 61.0 | 75.0 | 84.0 | 75.0 | 60.9 | 75.0 | 75.0 | 75.0 | 27.1 | 75.0 | 41.0 | 75.0 | 84.7 | 75.0 |
| CLIPDetection | 71.6 | 75.8 | 45.2 | 75.8 | 55.8 | 75.8 | 99.8 | 75.8 | 57.2 | 75.8 | 51.1 | 75.8 | 74.1 | 75.8 |
| AIDE | 82.5 | 98.6 | 84.2 | 98.6 | 6.0 | 98.6 | 90.9 | 98.6 | 94.0 | 98.6 | 1.5 | 98.6 | 79.3 | 98.6 |
| SAFE | 94.6 | 99.8 | 85.6 | 99.8 | 0.1 | 99.8 | 94.9 | 99.8 | 52.6 | 99.8 | 0.0 | 99.8 | 86.5 | 99.8 |
| PLM | 96.9 | 100.0 | 86.3 | 99.9 | 0.1 | 99.8 | 97.8 | 99.8 | 96.3 | 99.8 | 0.1 | 99.9 | 99.0 | 99.8 |
| FSD (10-shot) | 65.0 | 90.3 | 64.3 | 94.3 | 63.7 | 90.7 | 76.0 | 83.0 | 53.3 | 52.7 | 62.0 | 87.7 | 51.0 | 50.7 |
| **Ours (10-shot)** | 76.4 | 98.9 | 82.7 | 99.2 | 93.4 | 97.4 | 99.9 | 99.9 | 77.2 | 97.4 | 87.0 | 98.6 | 90.9 | 92.3 |

| Method | GLIDE | | GPT-4o | | GigaGAN | | HiDream-I1-Dev | | HunyuanDiT | | HunyuanImage 3.0 | | Imagen | |
|---|---|---|---|---|---|---|---|---|---|---|---|---|---|---|
| | AI | Non-AI | AI | Non-AI | AI | Non-AI | AI | Non-AI | AI | Non-AI | AI | Non-AI | AI | Non-AI |
| FreqNet | 80.8 | 75.0 | 83.8 | 75.0 | 98.5 | 75.0 | 98.6 | 75.0 | 67.9 | 75.0 | 88.2 | 75.0 | 90.3 | 75.0 |
| CLIPDetection | 81.5 | 75.8 | 70.5 | 75.8 | 79.5 | 75.8 | 84.1 | 75.8 | 80.8 | 75.8 | 86.8 | 75.8 | 89.9 | 75.8 |
| AIDE | 98.6 | 98.6 | 75.2 | 98.6 | 97.6 | 98.6 | 99.9 | 98.6 | 75.2 | 98.6 | 99.5 | 98.6 | 96.1 | 98.6 |
| SAFE | 95.0 | 99.8 | 88.1 | 99.8 | 92.5 | 99.8 | 99.5 | 99.8 | 93.5 | 99.8 | 96.7 | 99.8 | 54.9 | 99.8 |
| PLM | 99.7 | 99.8 | 90.8 | 99.9 | 99.3 | 99.8 | 99.9 | 99.8 | 96.9 | 99.9 | 98.6 | 99.7 | 86.5 | 99.9 |
| FSD (10-shot) | 68.3 | 85.3 | 59.3 | 71.7 | 59.0 | 75.3 | 76.7 | 93.3 | 68.0 | 88.0 | 60.3 | 73.3 | 85.0 | 96.7 |
| **Ours (10-shot)** | 88.0 | 99.3 | 90.0 | 94.5 | 97.0 | 98.5 | 88.4 | 97.5 | 89.9 | 99.0 | 96.9 | 98.4 | 98.1 | 98.5 |

| Method | Imagen 4 | | Infinity | | Janus-Pro-7B | | Kolors | | LlamaGen | | LongCat-Image | | LUMINA-Image 2.0 | |
|---|---|---|---|---|---|---|---|---|---|---|---|---|---|---|
| | AI | Non-AI | AI | Non-AI | AI | Non-AI | AI | Non-AI | AI | Non-AI | AI | Non-AI | AI | Non-AI |
| FreqNet | 67.7 | 75.0 | 94.2 | 75.0 | 98.9 | 75.0 | 91.8 | 75.0 | 86.7 | 75.0 | 48.7 | 75.0 | 79.1 | 75.0 |
| CLIPDetection | 73.8 | 75.8 | 80.5 | 75.8 | 85.8 | 75.8 | 81.2 | 75.8 | 89.9 | 75.8 | 86.9 | 75.8 | 85.7 | 75.8 |
| AIDE | 16.2 | 98.6 | 97.2 | 98.6 | 97.3 | 98.6 | 92.3 | 98.6 | 94.5 | 98.6 | 87.3 | 98.6 | 84.2 | 98.6 |
| SAFE | 0.6 | 99.8 | 99.9 | 99.8 | 99.8 | 99.8 | 98.4 | 99.8 | 99.0 | 99.8 | 95.4 | 99.8 | 90.4 | 99.8 |
| PLM | 0.1 | 99.8 | 98.7 | 99.8 | 99.2 | 99.8 | 99.2 | 99.8 | 99.2 | 99.8 | 96.1 | 100.0 | 96.1 | 99.8 |
| FSD (10-shot) | 58.7 | 79.7 | 69.7 | 92.3 | 73.3 | 93.7 | 71.3 | 93.3 | 57.3 | 84.3 | 52.7 | 75.7 | 66.7 | 87.0 |
| **Ours (10-shot)** | 91.3 | 94.2 | 86.7 | 98.6 | 98.2 | 99.6 | 99.3 | 99.2 | 99.5 | 99.6 | 89.6 | 96.7 | 94.1 | 99.0 |

| Method | MAE | | Midjourney V6.1 | | Midjourney V7 | | Midjourney V4 | | Midjourney V5 | | Midjourney V6 | | Nano Banana | |
|---|---|---|---|---|---|---|---|---|---|---|---|---|---|---|
| | AI | Non-AI | AI | Non-AI | AI | Non-AI | AI | Non-AI | AI | Non-AI | AI | Non-AI | AI | Non-AI |
| FreqNet | 84.2 | 75.0 | 35.1 | 75.0 | 26.1 | 75.0 | 81.8 | 75.0 | 81.0 | 75.0 | 67.0 | 75.0 | 81.0 | 75.0 |
| CLIPDetection | 70.0 | 75.8 | 51.2 | 75.8 | 41.7 | 75.8 | 67.3 | 75.8 | 60.6 | 75.8 | 65.5 | 75.8 | 73.6 | 75.8 |
| AIDE | 97.9 | 98.6 | 1.0 | 98.6 | 1.0 | 98.6 | 71.7 | 98.6 | 63.9 | 98.6 | 69.5 | 98.6 | 90.1 | 98.6 |
| SAFE | 100.0 | 99.8 | 0.2 | 99.8 | 0.3 | 99.8 | 85.3 | 99.8 | 90.7 | 99.8 | 89.1 | 99.8 | 97.3 | 99.8 |
| PLM | 96.9 | 99.8 | 0.1 | 99.8 | 0.3 | 99.8 | 96.0 | 99.8 | 97.5 | 99.8 | 97.2 | 99.8 | 98.6 | 99.8 |
| FSD (10-shot) | 94.3 | 98.7 | 56.3 | 76.0 | 49.3 | 64.0 | 58.0 | 90.7 | 71.3 | 91.0 | 51.3 | 69.0 | 57.0 | 75.0 |
| **Ours (10-shot)** | 99.0 | 99.5 | 62.4 | 98.0 | 52.0 | 94.9 | 95.6 | 99.1 | 79.2 | 96.4 | 92.8 | 96.7 | 78.2 | 97.3 |

| Method | Nano Banana Pro | | NextStep | | OmniGen | | OmniGen2 | | Playground V2 | | Playground V2.5 | | ProGAN | |
|---|---|---|---|---|---|---|---|---|---|---|---|---|---|---|
| | AI | Non-AI | AI | Non-AI | AI | Non-AI | AI | Non-AI | AI | Non-AI | AI | Non-AI | AI | Non-AI |
| FreqNet | 57.6 | 75.0 | 99.3 | 75.0 | 84.2 | 75.0 | 61.7 | 75.0 | 92.5 | 75.0 | 87.0 | 75.0 | 100.0 | 75.0 |
| CLIPDetection | 56.6 | 75.8 | 87.1 | 75.8 | 84.8 | 75.8 | 72.7 | 75.8 | 81.4 | 75.8 | 75.1 | 75.8 | 98.8 | 75.8 |
| AIDE | 4.5 | 98.6 | 97.4 | 98.6 | 79.5 | 98.6 | 83.1 | 98.6 | 95.0 | 98.6 | 89.5 | 98.6 | 99.5 | 98.6 |
| SAFE | 0.1 | 99.8 | 100.0 | 99.8 | 97.7 | 99.8 | 94.8 | 99.8 | 97.3 | 99.8 | 97.4 | 99.8 | 100.0 | 99.8 |
| PLM | 0.0 | 99.7 | 98.3 | 99.8 | 98.5 | 99.9 | 96.5 | 99.8 | 99.2 | 99.8 | 99.4 | 100.0 | 99.0 | 99.8 |
| FSD (10-shot) | 56.0 | 76.7 | 77.0 | 94.0 | 62.3 | 91.7 | 63.3 | 89.7 | 61.3 | 87.0 | 55.3 | 77.0 | 99.7 | 100.0 |
| **Ours (10-shot)** | 80.1 | 79.0 | 93.5 | 97.2 | 97.7 | 97.8 | 80.3 | 98.4 | 99.2 | 99.5 | 94.7 | 99.5 | 100.0 | 99.9 |

| Method | Qwen-Image | | SD3-Medium | | SDXL | | SD v1.4 | | SD v1.5 | | SD v2.1 | | SANA v1.5 | |
|---|---|---|---|---|---|---|---|---|---|---|---|---|---|---|
| | AI | Non-AI | AI | Non-AI | AI | Non-AI | AI | Non-AI | AI | Non-AI | AI | Non-AI | AI | Non-AI |
| FreqNet | 63.3 | 75.0 | 84.3 | 75.0 | 47.3 | 75.0 | 99.9 | 75.0 | 99.8 | 75.0 | 89.8 | 75.0 | 61.4 | 75.0 |
| CLIPDetection | 75.8 | 75.8 | 75.6 | 75.8 | 74.2 | 75.8 | 94.2 | 75.8 | 93.8 | 75.8 | 68.3 | 75.8 | 73.6 | 75.8 |
| AIDE | 95.9 | 98.6 | 80.5 | 98.6 | 93.8 | 98.6 | 99.9 | 98.6 | 99.7 | 98.6 | 97.6 | 98.6 | 95.8 | 98.6 |
| SAFE | 98.6 | 99.8 | 83.3 | 99.8 | 98.6 | 99.8 | 99.9 | 99.8 | 99.8 | 99.8 | 96.2 | 99.8 | 99.3 | 99.8 |
| PLM | 99.5 | 99.8 | 95.7 | 99.7 | 99.8 | 99.8 | 98.7 | 99.8 | 98.1 | 99.8 | 96.8 | 99.8 | 97.5 | 99.9 |
| FSD (10-shot) | 69.3 | 89.7 | 69.0 | 92.0 | 53.3 | 73.7 | 99.3 | 98.7 | 98.7 | 98.7 | 88.3 | 97.0 | 57.3 | 83.3 |
| **Ours (10-shot)** | 87.2 | 97.7 | 92.0 | 97.6 | 91.5 | 96.6 | 100.0 | 99.9 | 100.0 | 99.9 | 88.5 | 98.2 | 98.2 | 99.3 |

| Method | Show-o | | Show-o2 | | StarGAN | | StyleGAN3 | | VQDM | | Wukong | | Z-Image-Turbo | |
|---|---|---|---|---|---|---|---|---|---|---|---|---|---|---|
| | AI | Non-AI | AI | Non-AI | AI | Non-AI | AI | Non-AI | AI | Non-AI | AI | Non-AI | AI | Non-AI |
| FreqNet | 93.2 | 75.0 | 98.3 | 75.0 | 42.7 | 75.0 | 16.2 | 75.0 | 34.6 | 75.0 | 98.1 | 75.0 | 82.4 | 75.0 |
| CLIPDetection | 83.7 | 75.8 | 90.1 | 75.8 | 62.4 | 75.8 | 31.5 | 75.8 | 46.6 | 75.8 | 78.6 | 75.8 | 64.3 | 75.8 |
| AIDE | 89.4 | 98.6 | 98.0 | 98.6 | 35.5 | 98.6 | 3.7 | 98.6 | 89.8 | 98.6 | 98.8 | 98.6 | 94.5 | 98.6 |
| SAFE | 99.6 | 99.8 | 99.8 | 99.8 | 35.2 | 99.8 | 0.0 | 99.8 | 84.9 | 99.8 | 99.6 | 99.8 | 90.3 | 99.8 |
| PLM | 99.2 | 99.8 | 99.5 | 99.8 | 34.5 | 99.8 | 0.0 | 99.8 | 98.0 | 99.8 | 98.1 | 99.8 | 97.8 | 99.8 |
| FSD (10-shot) | 73.0 | 94.7 | 93.3 | 97.3 | 89.3 | 90.0 | 88.3 | 77.3 | 73.0 | 48.0 | 93.7 | 98.0 | 61.7 | 84.7 |
| **Ours (10-shot)** | 98.3 | 99.1 | 100.0 | 99.3 | 84.9 | 98.9 | 77.7 | 99.1 | 92.9 | 99.4 | 99.9 | 99.3 | 66.8 | 93.3 |

| Method | Doubao Seedream 3.0 | | Doubao Seedream 4.0 | | GPT-image-1.5 | | ideogram | | Ovis-U1 | | PixArt-$\alpha$ | | Sora-image | |
|---|---|---|---|---|---|---|---|---|---|---|---|---|---|---|
| | AI | Non-AI | AI | Non-AI | AI | Non-AI | AI | Non-AI | AI | Non-AI | AI | Non-AI | AI | Non-AI |
| FreqNet | 33.3 | 75.0 | 45.8 | 75.0 | 72.3 | 75.0 | 48.9 | 75.0 | 74.6 | 75.0 | 60.6 | 75.0 | 73.9 | 75.0 |
| CLIPDetection | 54.3 | 75.8 | 74.7 | 75.8 | 65.8 | 75.8 | 60.1 | 75.8 | 82.7 | 75.8 | 72.8 | 75.8 | 66.8 | 75.8 |
| AIDE | 0.4 | 98.6 | 0.0 | 98.6 | 75.7 | 98.6 | 2.1 | 98.6 | 78.6 | 98.6 | 41.3 | 98.6 | 75.2 | 98.6 |
| SAFE | 0.0 | 99.8 | 0.3 | 99.8 | 98.4 | 99.8 | 1.6 | 99.8 | 96.9 | 99.8 | 45.8 | 99.8 | 98.7 | 99.8 |
| PLM | 0.1 | 99.7 | 0.0 | 99.8 | 97.8 | 99.8 | 1.6 | 99.7 | 96.1 | 99.8 | 52.2 | 99.7 | 98.3 | 99.8 |
| FSD (10-shot) | 61.0 | 76.0 | 58.0 | 72.3 | 63.7 | 70.3 | 59.7 | 87.7 | 57.3 | 86.7 | 49.0 | 67.0 | 57.0 | 72.7 |
| **Ours (10-shot)** | 80.0 | 89.3 | 73.1 | 93.0 | 81.2 | 87.7 | 80.3 | 97.2 | 96.7 | 98.9 | 63.3 | 89.4 | 61.3 | 92.5 |

| Method | wan2.2-t2i-flash | | wan2.5-t2i-preview | |
|---|---|---|---|---|
| | AI | Non-AI | AI | Non-AI |
| FreqNet | 76.6 | 75.0 | 66.7 | 75.0 |
| CLIPDetection | 86.9 | 75.8 | 83.9 | 75.8 |
| AIDE | 70.6 | 98.6 | 92.1 | 98.6 |
| SAFE | 98.2 | 99.8 | 99.5 | 99.8 |
| PLM | 76.5 | 99.7 | 56.4 | 99.8 |
| FSD (10-shot) | 78.3 | 94.3 | 63.0 | 81.3 |
| **Ours (10-shot)** | 83.5 | 96.9 | 95.9 | 92.1 |

*Table 13.* Few-shot performance on the Pretrain Validation set of Treasure.

| Setting | Method | ADM | | BAGEL | | BRIA v3.2 | | BigGAN | | CogView2 | | CogView4 | |
|---|---|---|---|---|---|---|---|---|---|---|---|---|---|
| | | AI | Non-AI | AI | Non-AI | AI | Non-AI | AI | Non-AI | AI | Non-AI | AI | Non-AI |
| FewShot | FSD (10-shot) | 35.1 | 27.2 | 69.5 | 97.5 | 76.7 | 97.4 | 64.1 | 65.8 | 94.6 | 97.8 | 71.2 | 89.9 |
| | **Ours (10-shot)** | 100.0 | 99.8 | 100.0 | 100.0 | 100.0 | 100.0 | 100.0 | 100.0 | 99.9 | 100.0 | 100.0 | 99.9 |

| Method | Cogview3plus | | DALL·E 2 | | DALL·E 3 | | DF-GAN | | DeepFloyd IF | | FLUX.1-dev | | FLUX.2 | |
|---|---|---|---|---|---|---|---|---|---|---|---|---|---|
| | AI | Non-AI | AI | Non-AI | AI | Non-AI | AI | Non-AI | AI | Non-AI | AI | Non-AI | AI | Non-AI |
| FSD (10-shot) | 93.0 | 95.9 | 61.7 | 98.0 | 58.7 | 97.2 | 88.5 | 75.4 | 69.3 | 48.5 | 58.4 | 96.4 | 51.5 | 55.5 |
| **Ours (10-shot)** | 100.0 | 100.0 | 100.0 | 100.0 | 100.0 | 100.0 | 99.9 | 100.0 | 100.0 | 100.0 | 100.0 | 100.0 | 100.0 | 100.0 |

| Method | GLIDE | | GPT-4o | | GigaGAN | | HiDream-I1-Dev | | HunyuanDiT | | HunyuanImage 3.0 | | Imagen | |
|---|---|---|---|---|---|---|---|---|---|---|---|---|---|
| | AI | Non-AI | AI | Non-AI | AI | Non-AI | AI | Non-AI | AI | Non-AI | AI | Non-AI | AI | Non-AI |
| FSD (10-shot) | 93.2 | 86.6 | 40.6 | 76.3 | 73.0 | 85.4 | 52.9 | 97.8 | 80.8 | 94.5 | 48.2 | 81.9 | 58.5 | 98.7 |
| **Ours (10-shot)** | 100.0 | 100.0 | 100.0 | 100.0 | 100.0 | 100.0 | 100.0 | 100.0 | 100.0 | 100.0 | 100.0 | 100.0 | 100.0 | 100.0 |

| Method | Imagen 4 | | Infinity | | Janus-Pro-7B | | Kolors | | LlamaGen | | LongCat-Image | | LUMINA-Image 2.0 | |
|---|---|---|---|---|---|---|---|---|---|---|---|---|---|
| | AI | Non-AI | AI | Non-AI | AI | Non-AI | AI | Non-AI | AI | Non-AI | AI | Non-AI | AI | Non-AI |
| FSD (10-shot) | 58.4 | 88.6 | 71.2 | 96.9 | 69.7 | 98.0 | 58.3 | 98.4 | 77.5 | 92.4 | 65.7 | 86.4 | 87.7 | 93.8 |
| **Ours (10-shot)** | 100.0 | 99.9 | 100.0 | 100.0 | 100.0 | 100.0 | 100.0 | 100.0 | 100.0 | 100.0 | 100.0 | 100.0 | 100.0 | 100.0 |

| Method | MAE | | Midjourney V6.1 | | Midjourney V7 | | Midjourney V4 | | Midjourney V5 | | Midjourney V6 | | Nano Banana | |
|---|---|---|---|---|---|---|---|---|---|---|---|---|---|
| | AI | Non-AI | AI | Non-AI | AI | Non-AI | AI | Non-AI | AI | Non-AI | AI | Non-AI | AI | Non-AI |
| FSD (10-shot) | 58.1 | 98.5 | 64.5 | 86.0 | 48.2 | 69.7 | 72.5 | 96.4 | 74.6 | 96.8 | 49.8 | 72.8 | 55.4 | 81.0 |
| **Ours (10-shot)** | 100.0 | 100.0 | 100.0 | 100.0 | 100.0 | 100.0 | 100.0 | 100.0 | 100.0 | 99.9 | 100.0 | 100.0 | 100.0 | 100.0 |

| Method | Nano Banana Pro | | NextStep | | OmniGen | | OmniGen2 | | Playground V2 | | Playground V2.5 | | ProGAN | |
|---|---|---|---|---|---|---|---|---|---|---|---|---|---|
| | AI | Non-AI | AI | Non-AI | AI | Non-AI | AI | Non-AI | AI | Non-AI | AI | Non-AI | AI | Non-AI |
| FSD (10-shot) | 60.0 | 83.0 | 85.3 | 97.9 | 74.2 | 97.4 | 87.9 | 95.7 | 68.0 | 93.9 | 69.5 | 82.9 | 49.9 | 99.9 |
| **Ours (10-shot)** | 100.0 | 99.3 | 100.0 | 99.9 | 100.0 | 100.0 | 100.0 | 100.0 | 100.0 | 100.0 | 100.0 | 100.0 | 99.9 | 100.0 |

| Method | Qwen-Image | | SD3-Medium | | SDXL | | SD v1.4 | | SD v1.5 | | SD v2.1 | | SANA v1.5 | |
|---|---|---|---|---|---|---|---|---|---|---|---|---|---|
| | AI | Non-AI | AI | Non-AI | AI | Non-AI | AI | Non-AI | AI | Non-AI | AI | Non-AI | AI | Non-AI |
| FSD (10-shot) | 55.9 | 94.7 | 75.9 | 97.0 | 51.6 | 81.9 | 50.2 | 99.8 | 50.3 | 99.8 | 54.0 | 99.0 | 70.0 | 90.3 |
| **Ours (10-shot)** | 100.0 | 100.0 | 100.0 | 100.0 | 100.0 | 100.0 | 99.9 | 100.0 | 99.9 | 100.0 | 100.0 | 99.9 | 100.0 | 100.0 |

| Method | Show-o | | Show-o2 | | StarGAN | | StyleGAN3 | | VQDM | | Wukong | | Z-Image-Turbo | |
|---|---|---|---|---|---|---|---|---|---|---|---|---|---|
| | AI | Non-AI | AI | Non-AI | AI | Non-AI | AI | Non-AI | AI | Non-AI | AI | Non-AI | AI | Non-AI |
| FSD (10-shot) | 55.3 | 98.4 | 54.8 | 99.2 | 69.5 | 88.9 | 48.7 | 57.5 | 27.9 | 32.1 | 51.5 | 99.5 | 64.4 | 93.1 |
| **Ours (10-shot)** | 100.0 | 100.0 | 100.0 | 100.0 | 100.0 | 99.9 | 100.0 | 99.8 | 100.0 | 100.0 | 100.0 | 100.0 | 100.0 | 100.0 |

| Method | Doubao Seedream 3.0 | | Doubao Seedream 4.0 | | GPT-image-1.5 | | ideogram | | Ovis-U1 | | PixArt-$\alpha$ | | Sora-image | |
|---|---|---|---|---|---|---|---|---|---|---|---|---|---|
| | AI | Non-AI | AI | Non-AI | AI | Non-AI | AI | Non-AI | AI | Non-AI | AI | Non-AI | AI | Non-AI |
| FSD (10-shot) | 51.4 | 83.0 | 55.4 | 78.1 | 53.6 | 79.5 | 61.3 | 93.8 | 82.0 | 94.2 | 49.4 | 68.1 | 53.5 | 79.9 |
| **Ours (10-shot)** | 100.0 | 99.9 | 100.0 | 100.0 | 100.0 | 100.0 | 100.0 | 100.0 | 100.0 | 100.0 | 100.0 | 99.9 | 100.0 | 100.0 |

| Method | wan2.2-t2i-flash | | wan2.5-t2i-preview | |
|---|---|---|---|---|
| | AI | Non-AI | AI | Non-AI |
| FSD (10-shot) | 56.2 | 98.3 | 61.9 | 90.1 |
| **Ours (10-shot)** | 100.0 | 100.0 | 100.0 | 100.0 |

