# OpenReview forum: "Fleet: Few Shots Lead Effective AI-generated Image Detection"
_ICML.cc/2026/Conference — ICML 2026 regular_

### Official Review · Reviewer_Tvk3 · 2026-02-25

**Soundness:** 3
**Presentation:** 4
**Significance:** 3
**Originality:** 3
**Overall Recommendation:** 5
**Confidence:** 5

**Summary:**

This paper introduces FLEET (Forensic Learning via Evolving Exemplar Tuning), a dynamic few-shot adaptation framework designed to effectively decouple real and fake features for AI-generated image detection. The model projects features extracted from a DINOv3 image encoder into multiple independent subspaces. Within these subspaces, it calculates routing weights by combining high-frequency signals with projected subspace addresses, which are then utilized to aggregate the final features. During the pre-training phase, an orthogonality constraint is applied to these routing weights, ensuring that specific subspaces are selectively activated depending on the authenticity of the input. Furthermore, during the few-shot adaptation process, FLEET employs an avoidance loss—formulated via dot product—to explicitly prevent novel fake data from inadvertently activating the routing weights associated with real images. Alongside the framework, this research proposes Treasure, a large-scale benchmark encompassing the latest AI generative technologies. Utilizing this dataset, the authors experimentally demonstrate the "Evaluation Blind Spot" of existing SOTA models, revealing that their high closed-set performance catastrophically fails to generalize against novel, unseen generative data.

**Compliance With Llm Reviewing Policy:**

Affirmed.

**Final Justification:**

My primary concerns centered on catastrophic forgetting in sequential adaptation settings, the structural distinction from conventional MoE, and the generalizability of the framework across different backbones. The authors' rebuttal effectively addressed the most critical of these: the additional 5-step sequential adaptation experiment is convincing and fully resolves my concern regarding catastrophic forgetting. The clarification on the MoE distinction and the rationale that high-frequency cues prevent semantic overfitting also make good sense — I encourage the authors to explicitly include these discussions in the camera-ready version.

For the remaining questions on backbone ablations and hard sample mining, I understand that these experiments were not feasible within the rebuttal period, and I appreciate the authors' commitment to addressing them in the final version.

Overall, the rebuttal has thoroughly addressed my core concerns, and I am happy to maintain my score of 5.

**Key Questions For Authors:**

1. Could the authors provide ablation studies using other popular vision foundation models (e.g., CLIP-ViT, Swin Transformer, ConvNeXt) to further validate the framework's generalizability beyond DINOv3 and Xception?

2. Could you clarify the exact data ratio of the Replay Set during few-shot adaptation (e.g., for a 1-shot task, is it exactly 500 Non-AI and 501 AI)?

3. Furthermore, instead of random selection, have the authors considered applying 'hard sample mining' to construct the Replay Set? Selecting boundary or hard-to-classify samples could potentially improve adaptation efficiency and better mitigate catastrophic forgetting.

**Limitations:**

yes

**Strengths And Weaknesses:**

**Strengths:**
- **Unveiling Evaluation Blind Spots via the Treasure Dataset:** The introduction of the new Treasure dataset, which comprehensively encompasses the latest generative models, effectively highlights the "evaluation blind spot" of current state-of-the-art (SOTA) detection models. The authors successfully demonstrate that existing SOTA methods severely overfit to their closed-set training distributions and fail to generalize to novel data.

- **Shifting Paradigm to Dynamic Adaptation:** Recognizing the adversarial, cat-and-mouse nature of the generative AI and detection fields, the paper's shift from static generalization to a dynamic, few-shot adaptation framework is highly appropriate and urgently needed in the current research landscape.

- **Innovative Feature Decoupling via Orthogonal Routing:** The approach of applying orthogonality to routing weights to strictly decouple the activation of real and fake features is a highly innovative and elegant solution.

**Weaknesses:**
- **Lack of Distinction from Mixture-of-Experts (MoE):** The proposed mechanism of utilizing multiple subspaces and routing weights operates on a protocol very similar to the traditional Mixture-of-Experts (MoE) architecture. While adapting the MoE concept to separate real and fake features is interesting, the paper lacks a deeper theoretical discussion on its structural distinctiveness or superiority over standard MoE baselines.

- **Scalability in Continual Learning Scenarios:** As the authors partially acknowledge in the limitations, Figure 6a shows that the routing weights shift dramatically after a single adaptation step. In real-world scenarios, a model must continuously adapt to multiple, sequential threats. It remains unverified whether the proposed framework can sustain this multi-step adaptation without suffering from catastrophic forgetting over the long term.

- **Insufficient Motivation for Utilizing High-Frequency Features:** The motivation for using high-frequency components to compute subspace addresses is overly simplistic. Beyond merely citing previous studies that utilized high-frequency signals to capture low-level artifacts, the paper should provide a more in-depth and specific rationale for how these signals uniquely contribute to the orthogonal routing mechanism.

- **Lack of Qualitative Analysis and Interpretability:** The paper would significantly benefit from qualitative visualizations (e.g., attention maps or feature visualizations) comparing success cases (true positives) and failure cases. Demonstrating exactly which visual artifacts trigger specific routing switches, or visually analyzing where avoidance routing fails, would greatly enhance the interpretability and persuasiveness of the model.

---

> ### Author Rebuttal · Authors · 2026-03-31
>
> We are grateful to the reviewers for their thoughtful and constructive feedback. We are pleased that the reviewers recognized the novel paradigm for AIGI detection we proposed, as well as our contributions regarding methodology and datasets. In response to the concerns raised by the reviewers, we offer the following seven-point response:
>
> **(1) Relation to MoE.**
>
> Fleet is structurally related to MoE because both use routing, but their goals differ. Classical MoE increases model capacity through expert collaboration and usually relies on load balancing. Fleet instead aims at **information isolation**: preventing AI-generated features, especially realistic ones, from contaminating the Non-AI space. Therefore, Fleet does not optimize balanced usage which is used in MoE; during adaptation, the avoidance-routing loss explicitly pushes novel AI samples away from the Non-AI anchor, enforcing subspace separation. We will clarify this distinction in the final version.
>
> **(2) On sequential adaptation.**
>
> We agree that Fig. 6a shows substantial routing redistribution after one adaptation step. However, this does not necessarily indicate destructive forgetting; it can reflect beneficial reallocation when adapting from pretrained generators (ProGAN/SD v1.4) to a new artifact distribution such as GPT-4o.
> To test this, we conducted a preliminary 5-step sequential adaptation experiment on unseen generators . After each step, we evaluated the current task, all previous tasks, and the original pretraining validation set. Results remain stable: pretraining-validation accuracy drops only from 99.98% to 99.78%, while previous tasks remain stable or slightly improve (e.g. CogView4: 83.55% → 84.10%). This suggests that routing redistribution does not lead to catastrophic forgetting in our tested setting. Detailed results are shown below.
> ||Pretrain-val|StarGAN|SDXL|GPT-4o|CogView4|Seedream4.0|
> |---|---|---|---|---|---|---|
> |After Pretrain|99.98|-|-|-|-|-|
> |After StarGAN|99.97|70.70→93.55|-|-|-|-|
> |After SDXL|99.87|93.35|83.45→95.35|-|-|-|
> |After GPT-4o|99.84|93.50|95.05|70.45→87.35|-|-|
> |After CogView4|99.86|94.30|94.40|87.50|79.10→83.55|-|
> |After Seedream4.0|99.78|93.45|94.95|91.05|84.10|78.15→84.90|
>
> **(3) On high-frequency-guided routing.**
>
> Thank you for noting that our motivation was underexplained. In Fleet, high-frequency information does not replace the pretrained visual backbone or act as the main discriminative source. Instead, it provides a routing cue that is less affected by semantics and more sensitive to generation artifacts.
> This is especially useful in few-shot adaptation. Since the support set is very small, routing dominated by semantic features can easily encode content bias from support images and cause semantic overfitting. High-frequency cues are better suited to decide which subspace the information should be sent to. Table 2 supports this: removing frequency-guided routing reduces mAcc from 93.26% to 89.91%.
>
> **(4) Qualitative analysis.**
>
> Following the reviewer’s suggestion, we collected success and failure cases and visualized their routing activations and attention heatmaps. Since the routing signal is derived from frequency information, the patterns do not always align cleanly with human semantic intuition, so we did not observe a simple visual rule explaining every routing decision. We provide some cases on https://imgur.com/a/jzJ8yrx.
>
> **(5) On other visual backbones.**
>
> We agree that more backbone experiments would better evaluate Fleet’s extensibility, and we will add them in the final version. Our choice of DINOv3 is not arbitrary: recent work [1] shows that modern self-supervised visual foundation models already provide strong AIGI detection generalization, partly because they preserve discriminative low-level cues better than semantically aligned models such as CLIP. For this reason, DINOv3 is a strong backbone for studying few-shot adaptation to emerging generators. We agree that adding other backbones will further clarify Fleet’s scope and robustness.
>
> **(6) On the few-shot setting.**
>
> During adaptation, training uses two parts: a replay set with 500 AI-generated and 500 Non-AI images sampled from pretraining data, and a support set. Under 10-shot, the support set contains 10 AI images from the new generator and 10 Non-AI images for class balance. Thus, the total number of images used in adaptation is **1020**. More details are given in Appendix A.1.3.
>
> **(7) On hard-sample replay.**
>
> We appreciate this suggestion. Constructing the replay buffer with hard samples is a promising extension and aligns well with Fleet. Due to rebuttal time limits, we cannot include a complete hard-sample replay design and experiments at this stage, but we will add this as an important extension in the final version.
>
> [1] Zhou Y, He X, Lin K, et al. Simplicity Prevails: The Emergence of Generalizable AIGI Detection in Visual Foundation Models[J]. arXiv preprint arXiv:2602.01738, 2026.

---

> > ### Author Rebuttal · Reviewer_Tvk3 · 2026-04-03
> >
> > Thank you to the authors for the strong and highly effective rebuttal. The additional 5-step sequential adaptation experiment is very convincing and fully resolves my concerns regarding catastrophic forgetting. Furthermore, the structural distinction from traditional MoE and the rationale that high-frequency cues prevent semantic overfitting make perfect sense. Please ensure these discussions are explicitly included in the camera-ready version. I completely understand the time constraints regarding the additional backbone ablations and hard-sample mining, and committing to include them in the final version is perfectly acceptable. Overall, the rebuttal has thoroughly addressed my core questions and strengthened my confidence in this work. I happily maintain my score of 5.

---

> > > ### Author Response · Authors · 2026-04-06
> > >
> > > We sincerely thank you for your highly professional review and your strong validation of our "urgently needed" paradigm shift, our "highly innovative" orthogonal routing, and how our Treasure dataset unveils critical evaluation blind spots. We are very glad our rebuttal fully resolved your concerns, and as promised, we will carefully incorporate the additional discussions and experiments into the next version.

---

### Official Review · Reviewer_VBrs · 2026-03-09

**Soundness:** 2
**Presentation:** 3
**Significance:** 3
**Originality:** 2
**Overall Recommendation:** 3
**Confidence:** 4

**Summary:**

Addressing the critical pain point that the prevalent "static artifact hypothesis" in the current AIGI (AI-Generated Image) detection field often fails against rapidly iterating closed-source generative models, this paper exposes the limitations of static feature extraction in open-world defense and proposes a novel dynamic anti-forgery paradigm of continuous evolution.
Centered around this new paradigm, this paper makes two main substantial contributions:

1. Constructing a highly challenging, large-scale benchmark dataset, Treasure. This dataset comprises approximately 360,000 images and extensively covers 64 mainstream generative models, including 20 of the latest closed-source commercial engines. Furthermore, it introduces a multi-dimensional evaluation system incorporating semantics and artistic styles (e.g., photorealism, 2D illustration, 3D rendering), thereby elevating the realistic challenge of testing scenarios.

2. Proposing a few-shot dynamic adaptation detection framework named Fleet. Adopting a "Shunt-and-Isolate" design philosophy, this method prevents interference between novel and historical knowledge via a dual-branch architecture and a subspace routing mechanism. It leverages high-frequency feature signals of images to guide the feature, extracted by a pre-trained large vision model, to be mapped into orthogonal and mutually exclusive subspaces for effective decoupling. Building upon this foundation, Fleet introduces an Avoidance Routing strategy, combined with a distillation mechanism utilizing a Replay Buffer, to achieve efficient few-shot learning and overcome the catastrophic forgetting problem during continuous fine-tuning. Extensive evaluations on both Treasure and AIGIBench demonstrate its effective few-shot rapid adaptation capabilities.

**Compliance With Llm Reviewing Policy:**

Affirmed.

**Final Justification:**

While the second-round rebuttal further clarifies some issues (e.g., optimization interaction and long-horizon adaptation), I still find that the core issue regarding orthogonal subspace disentanglement remains insufficiently justified.
In addition, for Question 2, although the authors provide additional experiments suggesting that avoidance routing does not harm Non-AI accuracy, the rebuttal still lacks analysis on decision boundary effects and gradient interactions between the objectives. This leaves the underlying optimization dynamics insufficiently explained.
In particular, although the authors argue that disentanglement emerges from the combination of routing and regularization mechanisms, the explanation of how orthogonal subspaces effectively decouple highly entangled nonlinear representations is still not fully convincing. This component forms the foundation of the proposed method, and its lack of clear theoretical or empirical validation limits my confidence in the overall soundness.
Given that this issue lies at the core of the method rather than a peripheral detail, I maintain my original score.

**Key Questions For Authors:**

1. Regarding the Robustness Experiment, the model's performance exhibits a noticeable decline on low-quality images. However, there is a lack of analysis concerning the source of this degradation—specifically, whether the impact primarily stems from the High-Frequency Component-Guided Subspace Routing or the subsequent model feature extraction process. Furthermore, the paper lacks a sound theoretical justification or analytical discussion demonstrating that the proposed method remains reliable under conditions of degraded image quality.

2. Regarding the forced routing fine-tuning for "subtle AI features that closely resemble authentic images" (as mentioned in Weakness 2), does this fundamentally exacerbate the model's confusion regarding original authentic (Non-AI) images, thereby creating a fundamental mathematical conflict with the Anti-Forgetting Distillation objective? We request the authors to provide a discussion or conduct relevant experiments on the update dynamics between the Avoidance Loss and the Distillation Loss, particularly in the marginal regions of hard-to-distinguish authentic images. Specifically, if such a significant conflict exists, the authors should clarify why the framework achieves stable adaptation and analyze whether there is an inherent "self-contradiction" among the optimization objectives.

3. It is highly recommended that the authors conduct preliminary experiments on long-sequence continual adaptation involving at least multiple novel generators. This would serve to verify the representational capacity limits of a fixed number of subspaces and the overall performance stability of the model following multiple successive fine-tuning phases. Specific aspects to explore include potential feature collisions among low-dimensional subspaces and the decoupling efficacy of linear projections when handling complex features.

4. In practice, the paper relies heavily on the Replay Buffer to compute the distillation loss, and the dynamic updating of this buffer requires a substantial amount of historical data. This renders the proposed method more akin to "replay-based continual learning" rather than "few-shot adaptation" in the strict sense. Under long-sequence continual adaptation, would this framework still be able to maintain its current adaptation efficiency if historical storage of multiple unseen image types were unavailable?

5. Considering that pre-trained vision models such as DINOv3 encode semantic and textural features into the same manifold in a deeply non-linear manner, is employing a simple global affine transformation sufficient as a decoupling mechanism to effectively isolate independent artifact distributions from such highly abstract semantic manifolds? The authors are requested to discuss whether this architecture faces the risk of Representation Degradation and to clarify its capability to effectively resolve this issue.

**Limitations:**

Yes

**Strengths And Weaknesses:**

Strengths:

1. This paper accurately captures the critical pain points in the field: existing datasets and detection model designs suffer from catastrophic performance collapse when confronted with rapidly iterating closed-source commercial large models. Shifting the detection paradigm from "static generalization" to "few-shot dynamic adaptation" represents a reliable and promising approach. The constructed Treasure dataset incorporates generated results from 20 closed-source commercial APIs; once open-sourced, it will bridge the gap in current AIGI detection evaluation benchmarks, significantly benefiting subsequent research in this domain.
2. The Fleet framework proposes improvements targeting several pain points in the current AIGI detection dilemma. It leverages the "Shunt-and-Isolate" philosophy to achieve precise identification and updating of image features; Avoidance Loss effectively guarantees the routing allocation for unseen AI images, while the introduction of Anti-Forgetting Distillation ensures the maintenance of performance during dynamic adaptation.
3. The proposed framework demonstrates excellent performance in practical scenarios involving single-class few-shot updates. Compared with current mainstream SOTA models, the proposed method achieves superior results, indicating its practical significance.

Weaknesses:

1. The core routing mechanism of Fleet relies on the high-frequency signals of images to guide features, which imposes strict requirements on the quality of input images. According to the results of the robustness experiment, it can be observed that compared to traditional methods, the accuracy drops significantly when facing JPEG compression. Since image quality and high-frequency components cannot be guaranteed in real-world Internet environments, the efficacy of this method is substantially compromised.
2. Although Fleet targets multiple pain points, it overlooks the potential conflicts between different optimization objectives. While the framework is based on dynamic adaptation, its core routing mechanism remains highly dependent on traditional static high-frequency features. When considering the adaptation to unseen AI images, Avoidance Routing forcefully pushes generated images with inconspicuous high-frequency features into the AI subspace. Such forced fine-tuning is highly likely to cause the model to confuse the original decision boundaries between authentic and generated images, thereby creating a fundamental conflict with the objective of Anti-Forgetting Distillation. In practical implementation, the paper utilizes the Softmax function to compute routing weights for all subspaces; consequently, when faced with out-of-distribution unknown images, the model is forced to allocate them to relatively mismatched subspaces, which inevitably results in feature contamination.
3. The limits of the model's adaptability remain to be thoroughly investigated. The paper notes in its limitations that the current performance evaluation focuses solely on single-step few-shot adaptation. Under scenarios of long-term dynamic updating, where the model needs to adapt to multiple unseen generator models successively, the effectiveness of the subspaces for feature representation is questionable.
4. Regarding the few-shot adaptation capability, the model’s performance in practice is heavily reliant on the extensive dataset within the Replay Buffer, which deviates from the definition of a strictly "few-shot" adaptation. Furthermore, considering the dynamic adaptation requirement necessitates long-term, continuous updates to the Replay Buffer, the actual demand for images from subsequent unknown models is significantly higher than implied, making the few-shot performance claimed in the paper difficult to achieve in reality.
5. The feature decoupling mechanism of the framework is built merely upon basic affine transformations. In practical scenarios, AIGI artifacts exhibit deep non-linear semantic entanglement with the image context. Employing low-rank linear projection matrices to disentangle such high-order non-linear entanglement introduces a distinct Representation Bottleneck from a methodological standpoint. The paper lacks sufficient justification or empirical validation regarding the effectiveness and reliability of this mechanism.
6. There are several grammatical inaccuracies in the manuscript, such as "a mutually exclusive subspace routing methods" in Section 4.1 and "We analysis the impact..." in Section 5.4.

---

> ### Author Rebuttal · Authors · 2026-03-31
>
> We thank the reviewer for the constructive comments.
>
> **(1) On JPEG robustness and the source of degradation.**
>
> This is an important concern since it is common in practice. As shown in Fig. 5, Fleet is already more robust than the baselines under most JPEG compression levels. To further examine this issue, we conduct two additional analyses.
>
> 1. Under standard robustness training with JPEG-compressed images  at different quality levels added, Fleet maintains mAcc above 84% throughout, indicating practical usability under realistic compression.
>
> 2. By decoupling routing and feature extraction, we find that the main degradation comes from feature extraction rather than routing.
>
> Full results are provided on https://imgur.com/a/LSBmT8B and will be included in the revision.
>
> **(2) On possible conflict between avoidance routing and boundary preservation.**
>
> The key concern is whether avoidance routing may push highly realistic fake images into the AI subspace and thus harm the real-image boundary. To test this, we ran extra experiments on GPT-4o and Seedream 4.0 under three settings: (i) avoidance loss with classification disabled in the first five epochs, (ii) avoidance loss with classification enabled throughout, and (iii) no avoidance loss. **Adding avoidance loss clearly improves AI accuracy, while Non-AI accuracy remains nearly unchanged.** This suggests that avoidance routing helps separate subtle fake samples from the real-image region without damaging the real-image boundary. We further reduce optimization conflict through staged training: routing is stabilized first, then the decision boundary is refined. Routing weight allocations and loss curves are shown on https://imgur.com/a/7YV1l49 .
>
> **(3) On continual adaptation.**
>
> We agree that the capacity of a fixed number of subspaces under long adaptation sequences is important. Our paper focuses on the practically critical problem of early-stage few-shot fast adaptation rather than long-horizon continual learning. To partially address this concern, we conducted a 5-step sequential adaptation experiment on five unseen generators. After each step, we evaluated the current task, all previous tasks, and the pretraining validation set. We did not observe meaningful degradation: pretraining-validation accuracy drops only from 99.98% to 99.78%, while previous tasks remain stable or slightly improve (e.g., GPT-4o: 87.35% → 91.05%). These results suggest that Fleet does not show clear subspace collision or loss of disentangling ability in the tested setting. Detail results are shown on https://imgur.com/a/N3LZNUG . We will extend this analysis to longer sequences in the final version.
>
> **(4) On replay and the meaning of “few-shot.”**
>
> We would like to clarify that **"few-shot" specifically refers to the amount of supervision required from the new generator**. The task setting is **driven by practical deployment needs**, rather than by mechanically following the strict classical definition of few-shot learning. In practice, the scarce resource is data from the new generator, whereas maintaining a small historical buffer(1000 samples) is realistic and operationally affordable. From the perspective of research, our setting is closer to the **intersection of few-shot learning and continual learning**: the model must adapt to a new generator with minimal samples while preserving previously acquired capability.
>
> **(5) On whether affine projection is sufficient.**
>
> Our claim is not that a single affine transform alone can fully disentangle the nonlinear VFM feature manifold. In Fleet, disentanglement comes from the combination of mechanisms: frequency-guided routing provides generator-sensitive cues, orthogonality promotes subspace specialization, and avoidance routing suppresses undesirable activation. The affine layers are only a lightweight parameterization of subspaces, not a standalone disentangling module. This design is intentional: we aim to preserve pretrained representations while extracting the most detection-relevant cues. This is also consistent with prior AIGI detection work such as Effort[1] and VIB-Net[2], showing that lightweight bottlenecks or orthogonal subspace modeling can effectively isolate forgery-related information without heavy nonlinear remapping.
>
> [1] Yan Z, Wang J, Jin P, et al. Orthogonal subspace decomposition for generalizable ai-generated image detection[J]. arXiv preprint arXiv:2411.15633, 2024.
>
> [2] Zhang H, He Q, Bi X, et al. Towards universal ai-generated image detection by variational information bottleneck network[C]//Proceedings of the Computer Vision and Pattern Recognition Conference. 2025: 23828-23837.

---

> > ### Author Rebuttal · Reviewer_VBrs · 2026-04-03
> >
> > Thank you for the detailed and thoughtful rebuttal. The additional analyses and experiments have improved the clarity of the paper and addressed several of my concerns.
> >
> > Question 1
> > The additional experiments and the decoupling analysis identifying feature extraction as the main source of degradation are convincing. This concern is sufficiently addressed.
> >
> > Question 2
> > The additional visualizations demonstrate that avoidance routing leads to cleaner subspace allocation, which is helpful for understanding the routing behavior. However, the response lacks analysis addressing the core mechanism.
> > In particular, it would be valuable to further clarify whether these objectives introduce competing gradients or affect decision boundaries, especially for hard or near-boundary samples.
> >
> > Question 3
> > The newly added 5-step sequential adaptation experiment provides useful evidence that the framework maintains stable performance on previously learned tasks across multiple updates. This is a positive step toward practical deployment and largely addresses my concern regarding short-term scalability.
> >
> > Question 4
> > I appreciate the clarification that “few-shot” refers to supervision from new generators, and the positioning of the method at the intersection of few-shot and continual learning is reasonable.
> > However, I still have a follow-up question regarding long-horizon scenarios: it would be helpful to further discuss how the framework behaves when earlier unseen generators may face limited replay data, and whether adaptation efficiency would degrade under such constraints.
> >
> > Question 5
> > The rebuttal clarifies that affine projections are not intended as a standalone disentangling mechanism, but rather part of a combined design. This explanation improves clarity.
> > At the same time, compared to the formulation in the paper where subspace projection is directly associated with feature decoupling, the rebuttal appears to place less emphasis on the role of affine projections. It would be helpful to further clarify this relationship and ensure consistency in how the role of affine transformations is characterized.

---

> > > ### Author Response · Authors · 2026-04-06
> > >
> > > Thank you for the helpful follow-up. We respond to Questions 2, 4, and 5 below.
> > >
> > > **Q2. On possible conflict between avoidance routing and boundary preservation.**
> > >
> > > These objectives are not directly opposed in Fleet. Avoidance loss acts on routing distributions of novel fake samples in the support set and suppresses their erroneous activation in the Non-AI subspace. Distillation loss acts on replay samples and preserves previously learned feature geometry. Thus, they do not impose opposite constraints on the same samples or the same representation target. Also, avoidance routing does not force samples toward a fixed fake prototype; it first corrects clearly wrong Non-AI activation, i.e., a local routing correction rather than a wholesale overwrite of the original boundary.
> > >
> > > Our additional analysis focuses on two difficult generators, **GPT-4o** and **Doubao Seedream 4.0**, both with very low zero-shot performance and thus more relevant to the reviewer’s concern about hard/near-boundary cases. Even there, adding avoidance loss consistently improves AI accuracy, while Non-AI and pretrain-validation performance remain nearly unchanged. If a substantial conflict existed, one would expect a clear trade-off—higher AI accuracy but lower Non-AI / pretrain-validation accuracy—or unstable oscillatory optimization. We observe neither. The loss curves shown at  https://imgur.com/a/DKleFYl  also do not show persistent opposite-direction oscillation: avoidance drops rapidly and converges, while distillation remains stable. This matches our staged training strategy: routing is corrected first, then boundary refinement is performed on a stabilized basis. Overall, avoidance routing helps separate subtle synthetic samples from the real-image region without harmful shifts of the real-image boundary.
> > >
> > > **Q4. On long-horizon sequential adaptation and limited replay.**
> > >
> > > We first clarify the scope of the paper. Our primary goal is **early-response few-shot adaptation**: rapidly recovering performance on an emerging generator from very limited samples while preserving prior capability. Concretely, we target the deployment gap between the release of a new generator and the point at which enough data are available for full-scale retraining or broader finetuning. This is already a practically important setting. The reviewer’s question concerns a broader long-horizon continual adaptation setting beyond the core scope of this paper, and should not redefine the main criterion for evaluating the present contribution.
> > >
> > > That said, to directly address this concern, we additionally conducted a **20-step sequential adaptation** experiment as a stress test beyond the paper’s main scope. As adaptation proceeds, the proportion of pretraining data in the replay buffer decreases continuously, making the setting progressively harder. Even so, pretraining validation remains nearly unchanged across all 20 stages, the first five early adapted categories stay stable, and each later stage still yields clear gains over both zero-shot and before-training performance. While later stages may partially benefit from accumulated exposure to more generators, the key conclusion is that Fleet does not show collapse in either retention or adaptation efficiency in this much longer sequential setting. In most practical deployments, the sequence length used in our main experiments is already sufficient for the realistic gap described above; the 20-step result is additional evidence that the framework remains stable even beyond that regime. Detailed results of the 20-step sequential adaptation experiment are provided at https://imgur.com/a/AQn6kG8 .
> > >
> > > **Q5. On the role of the projection layers.**
> > >
> > > We agree that the wording in the manuscript can be more precise. In Fleet, the projection layers map the high-dimensional backbone feature into learnable subspace representations, including subspace addresses and subspace features. Its role is to provide the parameterized subspace structure on which the rest of the framework operates. The effective decoupling is then induced jointly by the **subsequent mechanisms**: frequency-guided routing determines how inputs are allocated across subspaces, orthogonality and coverage constraints encourage these subspaces to develop distinct functional roles during pretraining, and avoidance routing further suppresses incorrect Non-AI activation during adaptation. Therefore, the projection layers are the structural basis for subspace modeling, while the effective separation emerges from the full routing-and-regularization process rather than from the projection layers alone.
> > >
> > > We will revise this description in the next version to explicitly distinguish **subspace parameterization** from the mechanisms that induce **functional decoupling**.

---

### Official Review · Reviewer_kZuP · 2026-03-09

**Soundness:** 4
**Presentation:** 4
**Significance:** 4
**Originality:** 4
**Overall Recommendation:** 5
**Confidence:** 4

**Summary:**

The paper proposes a dynamic paradigm of continuous few-shot evolution.

The paper proposes a dual-branch architecture named Fleet. The high-frequency branch guides subspace routing, while the pre-trained branch handles feature extraction and discrimination.

The authors construct the Treasure benchmark for AIGI detection. Treasure covers 64 models (including 20 closed-source commercial engines such as Nano Banana Pro and Imagen 4) and 360k images, and introduces multi-dimensional annotations of style and semantics.
The paper conducts experiments on multiple benchmarks and introduces an anti-forgetting evaluation.

**Compliance With Llm Reviewing Policy:**

Affirmed.

**Final Justification:**

The authors have effectively addressed all technical concerns raised during the initial review. The additional experiments significantly strengthen the paper’s claims regarding sequential forgetting and routing stability. I am therefore increasing my score to 5 and recommending the paper for acceptance.

**Key Questions For Authors:**

1. Will the model shift if multiple rounds of continuous evolution are performed? While the paper demonstrates 99.9% retention of the original pre-trained knowledge, it fails to showcase knowledge retention within the adapted sequence. This is a core issue regarding the viability of the dynamic paradigm in practical applications.

2. In formula (2), $A_i$ is obtained by linear projection of the input feature $z$: $$A_{i}=zW_{i}^{addr.}+b_{i}^{addr.}$$
If both $A_i$ (address) and $S_i$ (feature) are functions of $z$, then in route matching (formula 5), the dot product of $r_{freq}$ and an address that "changes with the input" is calculated. Typically, the expert address/prototype of a routing mechanism (such as MoE) is the network parameter (statically learned), but here $A_i$ changes dynamically with the input. Will this lead to extremely unstable route distribution in the early stages of training?

3. Have you compared dynamic ${A_i}$ to a baseline where ${A_i}$ is set as a static learnable prototype? Does the performance gain from dynamic ${A_i}$ outweigh the optimization instability it brings?

**Limitations:**

yes

**Strengths And Weaknesses:**

Strengths:
1. The construction of the large-scale benchmark Treasure is of great value for the community assessment.
2. The proposed dynamic paradigm of continuous few-shot evolution meets the practical needs of open-world defense.
3. The Fleet architecture is well-designed and technically sound.
4. The paper compares SOTA methods on multiple benchmarks with convincing results.

Weaknesses:
1. Although the paper emphasizes the efficiency of few-shot learning, it still relies on a replay set of 1,000 samples to address the problem of catastrophic forgetting.
2. The paper does not discuss the initialization strategy for the subspace parameters and its impact on the convergence rate. It is recommended that the authors provide supplementary information on whether a specific initialization method was used to promote early subspace differentiation, or provide ablation experiments showing how the convergence curve changes with initialization.
3. No ablation experiments were provided regarding the LoRA rank.

---

> ### Author Rebuttal · Authors · 2026-03-31
>
> We thank the reviewer for the constructive feedback and helpful technical suggestions.
>
> **(1) On replay efficiency.**
>
> We agree that replay is important for mitigating forgetting and that its effect depends on buffer size, as commonly observed in replay-based continual learning. However, this does not contradict the few-shot efficiency emphasized in our paper. These are different dimensions. In Fleet, replay is used only during adaptation, so it introduces no deployment-time computation or test-time overhead. It also does not require an extra training stage; replay samples are simply integrated into the same lightweight adaptation process. From a practical perspective, maintaining a limited historical buffer is far cheaper than recollecting large-scale data and re-finetuning whenever a new generator appears. We will clarify this distinction more explicitly in the final version.
>
> **(2) On initialization of subspace parameters.**
>
> Thank you for this suggestion. In our current implementation, Fleet already converges stably under the standard initialization used in the paper, and all reported results were obtained without any specially designed subspace initialization. To test sensitivity, we additionally evaluated a structured MoE-style warm start inspired by partial re-initialization [1], where subspace projections are initialized through a pretrained low-dimensional projection layer. This leads to only a marginal change in overall performance (Acc: 93.26% → 93.53%), while the pretraining-validation accuracy remains unchanged (99.95% → 99.95%). This indicates that Fleet is not particularly sensitive to initialization and does not rely on special initialization tricks. The full comparison is provided on https://imgur.com/a/mXI315d .
>
> **(3) On LoRA rank ablation.**
>
> Following the reviewer’s suggestion, we conducted an additional LoRA-rank ablation. The results show that rank = 8 gives the best overall trade-off, while performance differences across a reasonable range of ranks are small (±2.01%). This suggests that Fleet is not highly sensitive to this hyperparameter. Full experiment results are shown on https://imgur.com/a/mF2ddZK .
>
> **(4) On retention under sequential adaptation.**
>
> This question is indeed very important, and it is exactly one of the original motivations of our work. Our broader goal is to move AIGI detection from a static zero-shot paradigm to continuous evolution under sequentially emerging generators. The present work should be viewed as **a foundational step toward sequential adaptation**, rather than an endpoint. At the same time, this first adaptation step is **itself important: 1) Methodologically:** it answers several prerequisite questions for future settings: can a detector recover with only a few samples, can it do so without forgetting, and is the proposed mechanism stable and effective at all? 2) **Practically:** it addresses a critical deployment window in which a new generator has broken the existing detector, but sufficient data for full retraining is still unavailable.
>
> To further examine the broader goal, we additionally conducted a preliminary 5-step sequential adaptation experiment on unseen generators. The results are encouraging: after all 5 updates, the mAcc on the pretrain validation set remains at 99.78%(vs. 99.98% before adaptation), and results on previously adapted generators stay strong or even improve (e.g., GPT-4o: 87.4->91.1). This suggests that Fleet has potential beyond the single-step setting. We will include these results in the revision, and sequential adaptation will **be a main focus of our next work**. Detailed results are shown in https://imgur.com/a/N3LZNUG .
>
> **(5) On dynamic vs. static addresses, and possible routing instability.**
>
> We interpret the reviewer’s two questions as one issue: whether input-dependent dynamic addresses improve expressiveness at the cost of unstable routing in early training.
>
> First, dynamic addresses do improve adaptation. Replacing the input-dependent dynamic address ($A_i$) with a static learnable prototype for each subspace reduces overall accuracy from 93.26% to 92.21%, with the main drop coming from the AI class. This suggests that dynamic addresses are more effective for modeling artifact patterns of emerging generators.
>
> Second, Fleet does not rely on static addresses for stability. During pretraining, the $L_{orth}$ encourages distinct routing patterns for AI and Non-AI samples, while the $L_{cov}$ prevents routing collapse. During adaptation, Fleet further uses replay-based fixed anchors, and in the first five epochs prioritizes routing correction before enabling the classification term, reducing potential optimization conflict. We also provide training-loss curves showing smooth optimization rather than severe early oscillation.
>
> All full comparisons are provided on https://imgur.com/a/HqAJHg3 .
>
> [1] Drop-Upcycling: Training Sparse Mixture of Experts with Partial Re-Initialization, ICLR 2025.

---

> > ### Author Rebuttal · Reviewer_kZuP · 2026-04-03
> >
> > Thank the authors for their detailed responses and the inclusion of additional experiments. These efforts have successfully addressed all of my concerns. Therefore, I am happy to increase my score to 5.

---

> > > ### Author Response · Authors · 2026-04-06
> > >
> > > Thank you very much for your thoughtful review, constructive suggestions, and updated assessment. We greatly appreciate your recognition of both our methodology and the Treasure benchmark, and your professional feedback has helped us improve the work substantially. We will incorporate the key points you highlighted into the next version to make the paper more complete and rigorous. Thank you again for your support.

---

### Official Review · Reviewer_VWtW · 2026-03-17

**Soundness:** 2
**Presentation:** 3
**Significance:** 2
**Originality:** 3
**Overall Recommendation:** 4
**Confidence:** 3

**Summary:**

This paper proposes Forensic Learning via Evolving Exemplar Tuning (Fleet), a paradigm shift from "static generalization" to "dynamic adaptation". Fleet employs a dual-space orthogonal framework guided by high-frequency components to decouple features into frozen "Non-AI" and adaptable "Forgery" subspaces. Its avoidance routing mechanism executes a "Shunt-and-Isolate" strategy, steering novel artifacts to the forgery subspace while suppressing Non-AI activation. Additionally, addressing the lag in existing benchmarks, the authors introduce Treasure, a large-scale benchmark comprising 360k images across 64 models, including diverse closed-source engines. Experiments demonstrate the effectiveness of Fleet.

**Compliance With Llm Reviewing Policy:**

Affirmed.

**Final Justification:**

The authors have addressed most of my concerns, although I remain concerned about the zero-shot performance of the method. However, I acknowledge the importance of dynamic adaptation for detecting unseen detectors. Therefore, I have decided to increase my score.

**Key Questions For Authors:**

Please refer to the Weaknesses Part for questions.

**Limitations:**

yes

**Strengths And Weaknesses:**

**Strengths**
1. The dual-space orthogonal framework effectively separates frozen "Non-AI" and adaptable "Forgery" subspaces, successfully mitigating catastrophic forgetting during few-shot adaptation.
2. The proposed Treasure benchmark comprises 360k images across 64 models, including 20 closed-source engines. This offers a more realistic evaluation environment than existing benchmarks lagging behind current generative capabilities.
3. Experiments shows that Fleet significantly outperforms the few-shot baseline FSD and the zero-shot baselines.

**Weaknesses**
1. The 1-shot performance (76.08%) merely approaches the zero-shot SOTA PLM (76.56%). Fleet lacks a distinct advantage in strict zero-shot scenarios where no support samples are available.
2. Acquiring labeled samples from unseen generators is infeasible in real-world adversarial settings.
3. Experiments primarily validate single-step adaptation rather than sequential task streams, which limit the practical deployment of Fleet.
4. Performance degrades significantly with smaller replay buffers (e.g., −13.92% with 20 samples), indicating strong dependency on buffer size. This not only incurs storage costs but also necessitates additional few-shot adaptation tuning.

---

> ### Author Rebuttal · Authors · 2026-03-31
>
> We thank the reviewer for the thoughtful and constructive feedback. We are glad that the reviewer recognized our work on the methodology and dataset. Our responses to the reviewer's four concerns are as follows:
>
> **(1) On comparison with zero-shot SOTA.**
>
> We agree that comparison with strong zero-shot baselines is important. Meanwhile, we would like to emphasize that  Fleet is not intended for the strict zero-shot setting where no target samples are available. Our target scenario is **open-world early-response adaptation**: to rapidly adapt to an emerging generator from only a handful of samples while preserving prior capabilities. Under this goal, the most relevant criteria are: (i) how effectively a method converts extremely limited supervision into improved detection on the new generator, and (ii) whether this adaptation preserves performance on previously learned data. From this perspective, the results are encouraging: Fleet already approaches PLM at 1-shot, and its advantage becomes clear as a few more samples are provided, while no collapse in detection capability occurs on the pretrained validation set (For 10 shots: old: ProGAN: 99.97->99.97,SDv1.4: 99.97->99.95, new: Seedream4.0: 20.4->73.1, GPT4o: 29.9->92.3, MidjourneyV6: 68.1->94.8 ). We will revise the paper to make this evaluation target and its interpretation more explicit.
>
> **(2) On the practicality of obtaining labeled samples.**
>
> We would like to clarify that Fleet does not assume access to a large labeled dataset from the unseen generator. It only requires **a very small number of confirmed AI-generated samples** for adaptation (e.g. 10 shots) without precise generator labels. We believe this is a realistic weak-supervision assumption in practice, since such samples may come from public demos, surfaced suspicious cased, or a few manually verified examples. Under this setting, the key challenge is how to adapt effectively from minimal supervision while preserving prior capability, which is exactly the scenario Fleet targets.
>
> **(3) On continuous adaptation beyond a single step.**
>
> This question is indeed very important, and it is exactly one of the original motivations of our work. Our broader goal is to move AIGI detection from a static zero-shot paradigm to continuous evolution under sequentially emerging generators. The present work should be viewed as **a foundational step toward sequential adaptation**, rather than an endpoint. At the same time, this first adaptation step is **itself important**: 1) **Methodologically**: it answers several prerequisite questions for future settings: can a detector recover with only a few samples, can it do so without forgetting, and is the proposed mechanism stable and effective at all? 2) **Practically**: it addresses a critical deployment window in which a new generator has broken the existing detector, but sufficient data for full retraining is still unavailable.
>
> To further examine the broader goal, we additionally conducted a preliminary 5-step sequential adaptation experiment on unseen generators. The results are encouraging: after all 5 updates, the mAcc on the pretrain validation set remains at 99.78%(vs. 99.98% before adaptation), and results on previously adapted generators stay strong or even improve (e.g., GPT-4o: 87.4->91.1). This suggests that Fleet has potential beyond the single-step setting. We will include these results in the revision, and sequential adaptation will **be a main focus of our next work**. Detailed results are shown below.
> ||Pretrain-val|StarGAN|SDXL|GPT-4o|CogView4|Seedream4.0|
> |---|---|---|---|---|---|---|
> |After Pretrain|99.98|-|-|-|-|-|
> |After StarGAN|99.97|70.70→93.55|-|-|-|-|
> |After SDXL|99.87|93.35|83.45→95.35|-|-|-|
> |After GPT-4o|99.84|93.50|95.05|70.45→87.35|-|-|
> |After CogView4|99.86|94.30|94.40|87.50|79.10→83.55|-|
> |After Seedream4.0|99.78|93.45|94.95|91.05|84.10|78.15→84.90|
>
> **(4) On replay and buffer dependence.**
>
> We agree that replay is important for mitigating forgetting, and that its effect depends on the buffer size, which is also consistent with prior continual-learning literature. That said, we would like to clarify why this does not create a prohibitive burden in practice: i) the replay set is used only during adaptation, not during inference, so it introduces no persistent deployment-time computation and no test-time storage cost; ii) Replay does not require an additional training stage beyond few-shot adaptation; it is simply integrated into the same adaptation process. More broadly, we view replay as a pragmatic trade-off: compared with full retraining or full-data finetuning, maintaining a limited historical buffer is substantially more economical. We will clarify this cost-benefit trade-off more explicitly in the final version.

---

> > ### Author Rebuttal · Reviewer_VWtW · 2026-04-03
> >
> > Thank you for the detailed response. The authors have addressed most of my concerns, although I remain concerned about the zero-shot performance of the method. However, I acknowledge the importance of dynamic adaptation for detecting unseen detectors.

---

> > > ### Author Response · Authors · 2026-04-06
> > >
> > > Thank you again for the follow-up. We would like to clarify more directly that the remaining zero-shot concern is separate from the main objective of this paper. Fleet is proposed for rapid few-shot adaptation to emerging generators under open-world deployment, so strict zero-shot performance is **not** the primary target the method is designed to optimize. Evaluating Fleet mainly through a zero-shot criterion would therefore assess it under a different task definition from the one studied in the paper.
> > >
> > > Nevertheless, to directly address your concern and provide a comprehensive comparison, we have evaluated the zero-shot performance of Fleet alongside a few-shot adaptation of the state-of-the-art zero-shot method, **PLM [AAAI26]**.
> > > 1. Comparable Zero-Shot Baseline: In a purely zero-shot setting on the AI-generated images from Treasure, **Fleet** achieves an accuracy of **74.68%**, which is **highly comparable** to the **76.56%** achieved by **PLM**.
> > > 2. Superior Adaptation Efficiency: Furthermore, we extended PLM to a 10-shot setting by fine-tuning the pre-trained PLM model with 10 novel fake samples. Under this setup, **PLM** achieves **78.66%** accuracy. In stark contrast, **Fleet** reaches a significantly higher accuracy of **88.21%** under the same 10-shot condition.
> > >
> > > These results clearly demonstrate that while Fleet shares a similar zero-shot starting point with SOTA methods like PLM, its targeted design for few-shot learning yields a profoundly higher adaptation efficiency. Fleet can **leverage limited data much more effectively** than methods explicitly designed for zero-shot generalization.
> > >
> > > We will revise the next version to make this task positioning and evaluation logic more explicit.

---

### Decision · Program_Chairs · 2026-04-30

**Decision:**

Accept (regular)

**Comment:**

The paper proposes a few shot adaptation strategy for AI-generated image detection. The main initial concern of the reviewers were: zero-shot performance and experiments on single-step adaptation only. The authors clarified that the proposed approach is not designed for the zero-shot scenario. They have provided additional results on 5 and 20 step sequential adaptation, which convinced a majority of the reviewers. One reviewer was still unconvinced about the disentanglement into orthogonal sub-spaces. However, this cannot be considered a red flag, given the convincing empirical results. One issue that has not been addressed in this paper is possible domain shift in non-AI images (e.g., due to the changes in camera technology, etc.). Overall, the paper makes a solid contribution to ICML.